# Multi-fold increase in rainforests tipping risk beyond 1.5-2⁰C warming

Chandrakant Singh[1,2,3,*], Ruud van der Ent[4], Ingo Fetzer[1,2,5], Lan Wang-Erlandsson[1,2,5]

[1]Stockholm Resilience Centre, Stockholm University, Stockholm, Sweden
[2]Bolin Centre for Climate Research, Stockholm University, Stockholm, Sweden
[3]Department of Space, Earth and Environment, Chalmers University of Technology, Gothenburg, Sweden
[4]Department of Water Management, Faculty of Civil Engineering and Geosciences, Delft University of Technology, Delft, The Netherlands
[5]Potsdam Institute for Climate Impact Research, Potsdam, Germany

*Corresponding author; E-mail: chandrakant.singh@su.se, chandrakant.singh@chalmers.se

**ORCID**
Chandrakant Singh: http://orcid.org/0000-0001-9092-1855
Ruud van der Ent: https://orcid.org/0000-0001-5450-4333
Ingo Fetzer: http://orcid.org/0000-0001-7335-5679
Lan Wang-Erlandsson: http://orcid.org/0000-0002-7739-5069

**Abstract**. Tropical rainforests rely on their root systems to access moisture stored in soil during wet periods for use during dry periods. When this root-zone soil moisture is inadequate to sustain a forest ecosystem, they transition to a savanna-like state, losing their native structure and functions. Yet the influence of climate change on ecosystem's root-zone soil moisture storage and their impact on rainforest ecosystems remain uncertain. This study assesses the future state of rainforests and the risk of forest-to-savanna transitions in South America and Africa under four shared socioeconomic pathways (SSP1-2.6, SSP2-4.5, SSP3-7.0, and SSP5-8.5). Using a mass-balance-based empirical understanding of root zone storage capacity ($S_r$), defined as the maximum volume of root zone soil moisture per unit area accessible to vegetation's roots for transpiration, we project how rainforest ecosystems will respond to future climate changes. We find that under the end-of-the-21$^{st}$-century climate, nearly one-third of the total forest area will be influenced by climate change. As the climate warms, forests will require a larger $S_r$ than they do under the current climate to sustain their ecosystem structure and functions, making them more susceptible to water limitations. Furthermore, warming beyond 1.5-2⁰C will significantly elevate the risk of a forest-savanna transition. In the Amazon, the forest area at risk of such a transition grows by about 1.7-5.8 times in size compared to the immediate lower warming scenario (e.g., SSP2-4.5 compared to SSP1-2.6). In contrast, the risk growth in the Congo is less substantial, ranging from 0.7-1.7 times. These insights underscore the urgent need to limit the rise of global surface temperature below the Paris Agreement to conserve rainforest ecosystems and associated ecosystem services.

# 1   Introduction

Tropical rainforests in the Amazon and Congo basins are critical to the Earth system since they store and sequester a large amount of carbon, host vast biodiversity, and regulate the global water cycle (Malhi et al., 2014). However, these forests are under severe pressure from climate and land-use changes (Davidson et al., 2012; Lewis et al., 2015; Malhi et al., 2008). These changes result in decreased precipitation, increased seasonality, and higher atmospheric water demand (Malhi et al., 2014), leading to soil moisture deficits that inhibit plant growth (Singh et al., 2020; Wang-Erlandsson et al., 2022). Furthermore, projected increases in drought frequency, severity, and duration under future climate change (Dai, 2011; Liu et al., 2018) pose imminent threats to the capacity of rainforests to maintain their native ecological structure and functions (i.e., forest resilience) (Bauman et al., 2022; Grimm et al., 2013; Jones et al., 2009).

Under water-deficit conditions, rainforests adapt by investing in their root systems to gain better access to soil moisture necessary to maintain their structure and functions (Singh et al., 2020, 2022). At the same time, the availability of surplus moisture at shallow depths minimises the need for ecosystems to invest in extensive (deeper and lateral) root systems (Bruno et al., 2006). Furthermore, forest ecosystems adapt to climate change by optimising water distribution through mechanisms such as hydraulic redistribution (Liu et al., 2020; Oliveira et al., 2005), enhancing water-use efficiency by regulating stomatal conductance, and even shedding leaves (Wolfe et al., 2016) to minimise moisture loss (Barros et al., 2019; Brum et al., 2019; Lammertsma et al., 2011). Despite their critical role, the dynamic influence of climate change on vegetation's rooting structure and subsoil moisture is challenging to measure at the ecosystem scale (Fan et al., 2017). Thus, understanding how moisture from wet periods is stored, transmitted, and lost from the soil, as well as how it is accessed by vegetation during dry periods, is critical to the ecohydrology and resilience of terrestrial ecosystems under climate change.

However, such ecohydrological dynamics remain challenging to incorporate in Earth System Models (ESMs) (Lenton, 2011; Maslin and Austin, 2012; Valdes, 2011) – complex mathematical representations of Earth system processes and interactions across different biospheres. This limits the capacity of ESMs to simulate tipping points as an emergent property of the system (i.e., properties that emerge due to multiple interactions between several system components, and are not the property of an individual component) (Hirota et al., 2021; Reyer et al., 2015; Singh et al., 2023). This constraint is mainly due to our poor understanding of complex mechanisms governing the ecosystem, which are not well represented in ESMs. This includes a limited understanding of vegetation-climate feedbacks (Boulton et al., 2013, 2017; Chai et al., 2021), subsoil moisture availability (Cheng et al., 2017), ecosystem adaptation dynamics (Yuan et al., 2022), the response time of forest ecosystems to climate change perturbations, and assumptions about future (i.e., prescribed) land-use change (Hurtt et al., 2020) in the ESMs. Furthermore, in the Earth system, some interactions still remain largely unknown, thereby making the prediction of (abrupt) forest-to-savanna transition (referring to changes in the dense-canopy structure of forests to one that mimics an open-canopy structure similar to savanna; hereafter referred to as forest-savanna transition) challenging (Drijfhout et al., 2015; Hall et al., 2019; Koch et al., 2021).

To understand the extent of rainforest tipping risks, there is a need to assess and contrast the forest resilience consequences of low-emission and current commitment trajectories with the more commonly used high-emission scenario (Jehn et al., 2022). However, the risk of forest-savanna transitions under various possible climate future scenarios is relatively under-investigated. As a result of the conflicting findings and scenario-dependent uncertainties, the Intergovernmental Panel on Climate Change (IPCC) has only low confidence about the possible tipping of the Amazon forest by the end of the 21$^{st}$ century (Canadell et al., 2021). However, with mounting empirical evidence on how climate change influences rainforest ecosystems (Boulton et al., 2022; Küçük et al., 2022; Singh et al., 2020, 2022), the research on rainforest resilience loss has accelerated considerably in the recent decade (Ahlström et al., 2017; Huntingford et al., 2013). Yet, forest resilience is often assessed based on changes in forest carbon stocks (Huntingford et al., 2013; Parry et al., 2022) or precipitation (Hirota et al., 2011; Staal et al., 2020; Zemp et al., 2017); and rarely on the subsoil moisture availability of the ecosystem (Singh et al., 2022).

This study aims to assess the state of rainforests and the risk of a forest-savanna transition under the end of the 21$^{st}$-century climate based on an empirical understanding of ecosystems' root zone storage dynamics. For this, we use mass-balance derived root zone storage capacity ($S_r$) – representing the maximum amount of soil moisture vegetation can access for transpiration (Gao et al., 2014; Singh et al., 2020; Wang-Erlandsson et al., 2016). Our use of $S_r$ is grounded in its effectiveness in representing ecosystems' access to soil moisture and their ability to modify above-ground structures accordingly (de Boer-Euser et al., 2016; Singh et al., 2020; Stocker et al., 2023; Wang-Erlandsson et al., 2016). It should be noted that we refer to rainforest tipping as a forest-savanna transition 'risk' since the timing of such transitions depends on the stochastic fluctuations of other environmental factors, beyond just hydroclimate (e.g., fire, human influence, species composition) (Cole et al., 2014; Cooper et al., 2020; Higgins and Scheiter, 2012; Poorter et al., 2016). Therefore, to project if an ecosystem is a forest or has tipped to savanna in the future, we assume the hydroclimate projected by the end of the 21$^{st}$ century (i.e., 2086-2100) and ecosystem are in equilibrium. However, we do not account for the time required for ecosystems to reach their (long-term) equilibrium state, which previous studies suggest can take between 50-200 years after crossing the tipping point (Armstrong McKay et al., 2022).

## 2    Methodology

### 2.1    Study Area

This study focuses on forest ecosystems (i.e., excluding savanna/grassland and vegetation in human-influenced ecosystems) extending between 15ºN–35ºS for South America and Africa.

### 2.2    Data

This analysis uses both empirical and ESM-simulated datasets of precipitation and evaporation. Empirical datasets include remotely sensed and observation-corrected precipitation and evaporation time-series. Empirical precipitation estimates at daily timestep are obtained from the Climate Hazards Group InfraRed Precipitation

with Station data (CHIRPS; 0.25º resolution) (Funk et al., 2015). Furthermore, empirical evaporation is derived using an equally-weighted ensemble of three different datasets – (i) Breathing Earth System Simulator (BESS; 0.5º resolution) (Jiang and Ryu, 2016) (ii) Penman-Monteith-Leuning (PML; 0.5º resolution) (Zhang et al., 2016) and (iii) FLUXCOM-RS (0.083º resolution) (Jung et al., 2019) - at monthly timestep. Here, evaporation represents the sum of all evaporated moisture from the soil, open water and vegetation, including interception and transpiration. We only selected evaporation datasets free from biome-dependent parameterisation (such as plant function types, stomatal conductance, and maximum root allocation depth) and soil layer depth (represents maximum depth of moisture uptake). Ultimately, all evaporation datasets are bilinearly interpolated to 0.25º resolution and downscaled to daily timestep using ERA5 evaporation (0.25º resolution) estimates (Hersbach et al., 2020). All empirical datasets are obtained for 2001-2012.

We also obtained precipitation and evaporation estimates from 33 ESMs (from 22 different institutes), which includes CMIP6-historical and four SSP scenario simulations (SSP1-2.6 leads to approx. 1.3-2.4ºC warming; SSP2-4.5 corresponds to 2.1-3.5ºC warming and is closest to the current trajectory according to the nationally determined contributions (Anon, 2015); SSP3-7.0 around 2.8-4.6ºC warming; and SSP5-8.5 represents 3.3-5.7ºC warming; ºC warming represents an increase in mean global surface temperature change by the end of 21$^{st}$ century relative to 1850-1900 (IPCC, 2021) (Fig. 1; Table S1 and S2). The historical estimates are obtained at a monthly timestep for 2000-2014, and the estimates under different SSPs are obtained for 2086-2100. Though obtained estimates from different ESMs are at different spatial resolutions, we bilinearly interpolated them to 0.25º for this analysis.

Finally, to minimise the influence of human activity and non-forest land cover on the natural water cycle, we utilised land-cover data to remove pixels with such features from our analysis. We began by removing human-influenced and non-forest land cover, such as savanna, grasslands, and water bodies, from Globcover, a global land-cover classification dataset by the European Space Agency (ESA) at 300m resolution (GlobCover land-use map, 2022). We then performed majority interpolation to convert the dataset to a 0.25º resolution and to mask grid cells with less than 50% forest cover. This step ensured that only grid cells with over 50% forest cover were classified as forests for further analysis.

**2.3    Root zone storage capacity-based framework for projecting forest transitions**

Vegetation uptakes soil moisture from its roots; thus, the availability of root zone moisture is a key element that mediates the interaction between vegetation and climate (Brooks et al., 2015; Küçük et al., 2022; Rosas et al., 2019; Wang-Erlandsson et al., 2022). However, measuring soil- (such as texture and porosity) and root-characteristics (such as vertical and lateral extent and soil moisture uptake profiles) that influence access to subsoil moisture are challenging to measure at ecosystem scales (Bruno et al., 2006). Furthermore, land-system models tend to oversimplify the transfer and storage of water in root-zone due to insufficient knowledge about soil-vegetation-climate interactions (Albasha et al., 2015; Hildebrandt et al., 2016; Wang et al., 2004). In such cases, the mass-balance approach-based $S_r$ provides a tangible and comprehensive understanding of ecosystem

access to moisture stored in the soil (de Boer-Euser et al., 2016; Gao et al., 2014; McCormick et al., 2021; Stocker et al., 2023).

### 2.3.1 Estimating mass-balance derived root zone storage capacity ($S_r$)

Derived using the mass-balance approach, $S_r$ represents the maximum amount of soil moisture accessed by vegetation for transpiration (Singh et al., 2020; Wang-Erlandsson et al., 2016). This methodology calculates the maximum extent of soil moisture within the reach of plant roots, assuming that ecosystems do not invest in expanding their root-zone storage beyond what is necessary to bridge the maximum (accumulated) water-deficit experienced by the vegetation during dry periods (i.e., periods in which evaporation is greater than rainfall, irrespective of the seasons). This maximum annual accumulated water deficit ($D_{a,y}$) experienced by the ecosystem is calculated using daily precipitation and evaporation estimates (Appendix A1 and Fig. A1). Subsoil moisture beyond the reach of plant roots is primarily controlled by gravity-induced gradients (de Boer-Euser et al., 2016) and is not available for transpiration. The rationale is that any extensive investment (i.e., more than necessary) in root expansion would require carbon allocation and, thus, is inefficient from the perspective of the plants (Gao et al., 2014; Schenk, 2008). Since this approach does not rely on prior information about vegetation, soil, or land cover-based, by using empirical (observation-based) datasets (Appendix A1 and Fig. A1), we capture the dynamics of actual soil moisture available for the ecosystems (Wang-Erlandsson et al., 2016). The detailed methodology for calculating $S_r$ using precipitation and evaporation estimates is outlined in Appendix A1.

In this mass-balance approach, $S_r$ only represents a hydrological buffer essential for maintaining the ecosystem's structure and functions (Gao et al., 2014; Wang-Erlandsson et al., 2016). However, other biotic and abiotic factors, such as root morphology, soil depth, and geological formations, can physically restrict $S_r$ by limiting rooting depth, rooting structure, and the soil's water-holding capacity (Canadell et al., 1996; Jackson et al., 1996; Schenk and Jackson, 2002) (Appendix A2). Additionally, soil properties like porosity or field capacity could necessitate a deeper rooting strategy in different soil types (e.g., between sandy and clayey soil) to achieve a comparable level of $S_r$ to sustain the ecosystem under future climate (Kukal and Irmak, 2023). However, this study assesses the impact of future climate change on the ecosystem's hydrological regime, focusing on the changes to the ecosystem's equilibrium state. Therefore, the direct influence of soil and root characteristics under future climate change on $S_r$ (Appendix A2) and forest transitions falls outside our current scope.

### 2.3.2 Determining root zone storage capacity thresholds for forest transitions

A recent study by Singh et al. (2020) demonstrated that $S_r$ can effectively represent an ecosystem's above-ground state (i.e., whether it is a forest or savanna) and its level of water–stress, based on root-zone moisture availability. In this study, we refine their terminology from 'water-stressed state' to 'water-limited state' to more precisely describe the effects of changes in hydroclimatic conditions on forest and savanna ecosystems. They classified

these terrestrial ecosystem responses into four distinct categories based on the relationship between tree cover density and root zone storage capacity ($S_r$) (for a more detailed description, see Singh et al., 2020):

i. **Lowly water-limited forest:** Dense forests (>70% tree cover) that receive ample rainfall (with daily precipitation exceeding evaporation year-round; Singh et al., 2020) result in a very low $D_{a,y}$ (Appendix A1). In such an environment, the top layer of the soil remains consistently damp, allowing for efficient soil moisture uptake through shallow roots (<1m; $S_r$ and maximum rooting depth comparison in Singh et al., 2020), as vegetation typically utilises the shortest available pathway for moisture uptake (Bruno et al., 2006). Consequently, these forest ecosystems can sustain themselves with a low $S_r$ (<100 mm) (Singh et al., 2020).

ii. **Moderately water-limited forest:** Although these forests retain a dense structure (>65% tree cover), the increased precipitation seasonality (evaporation rates remain the same as before; Singh et al., 2020) leads to a relatively higher $D_{a,y}$ (Appendix A1). This necessitates greater investment in their rooting systems to access subsoil moisture for dry periods, with $S_r$ for these ecosystems ranging between 100-400 mm in South America and 100-350 mm in Africa (Singh et al., 2020). Notably, this enhanced below-ground investment does not compromise the above-ground ecosystem structure, as evidenced by the changes in ecosystem rooting structure relative to tree cover (Singh et al., 2020).

iii. **Highly water-limited forest:** With further increase in precipitation seasonality (even negligible precipitation during dry seasons) and duration of dry period, forests need to maximise their $S_r$ to sustain their structure (see Fig. S2 and S3 in Singh et al., 2020). Maximum rooting depths of these ecosystems can typically range between 15-20m (Singh et al., 2020). Maintaining ecosystems under these conditions is costly from a subsoil investment perspective (Schenk, 2008), with regions in South America and Africa showing $S_r$ values as high as 750 mm and 450 mm, respectively (Singh et al., 2020). Consequently, these values represent the upper limits beyond which forest ecosystems cannot further enhance their $S_r$ (Singh et al., 2020).

Possible mechanisms suggest that these trees adapt by shedding leaves to minimise moisture loss (Wolfe et al., 2016). However, this adaptation can reduce photosynthetic activity, leading to declines in root growth, and heightening the risk of mortality from hydraulic failures due to the unavailability of soil moisture at accessible depths (Guswa, 2008). Furthermore, the accumulation of dry leaves also perpetuates forest fires, thinning the ecosystem even further (tree cover can drop as low as 30%) (Nepstad et al., 1999; Singh et al., 2020). Although increased tree mortality reduces competition for water, enabling some trees to survive, the heightened risk of hydraulic failures and forest fires makes these ecosystems highly susceptible to transitioning to savanna (Anderegg et al., 2016; Oliveras and Malhi, 2016; Sperry and Love, 2015).

iv. **Savanna-grassland regime** (hereafter referred to as **savanna**)**:** These ecosystems, typically characterised by an open, grass-dominated structure (tree cover <40%), have both a lower water availability and demand (both precipitation and evaporation are lower than in forest ecosystems) (Ratnam et al., 2011; Singh et al., 2020). Thus, requiring a lower hydrological buffer to sustain their structure and

functions. For these ecosystems, $S_r$ values can be as low as 100 mm (Singh et al., 2020). Although tree species in this ecosystem can develop deep roots (extending up to 20m; see Fig. 2 and 3 in Singh et al., 2020), the majority of the root biomass is concentrated in the shallow soil layers (top 30–50 cm; shallow water uptake profile) (February and Higgins, 2010; Schenk, 2008). This strategy allows for competitive moisture uptake between trees and grass species (Nippert and Holdo, 2015). This also suggests that, for savanna, deeper roots don't always necessitate a high $S_r$ (Singh et al., 2020).

The difference in $S_r$ thresholds between both continents is due to the presence of water-use-efficient C4 grasses in Africa (Still et al., 2003), which reduces the competitiveness for moisture uptake between tree species and grasses – leading to a lesser need for extensive $S_r$ in the African forest ecosystem (Singh et al., 2020). Furthermore, these adaptation dynamics align with the alternative stable state theory (i.e., forest's stabilising feedback under hydroclimatic changes and tipping risk beyond certain hydroclimatic extremes) (Hirota et al., 2011), which makes $S_r$ more representative of the transient state of the ecosystem than precipitation (Singh et al., 2022). We, thus, use these mass-balance derived $S_r$ thresholds to project rainforest transitions and tipping risk under future climate change. A detailed description of how previous studies have projected rainforest tipping (Table S3), and how $S_r$-based framework builds upon their shortcomings is mentioned in the Supplement.

### 2.3.3 Projecting forest transitions under future climate change

Due to the lack of appropriate metrics for vegetation structure (e.g., tree cover density, tree height, floristic patterns) and the reliance on assumptions about future land-use change (i.e., prescribed rather than biophysically simulated) in ESMs (Hurtt et al., 2020), we use hydroclimate from ESMs as a proxy to project forest transitions under future climate conditions. Using this proxy, we assume that the hydroclimate projected for the end of the 21st century and the ecosystem are in equilibrium (Staal et al., 2020). We start by classifying forests under the current climate following the approach by Singh et al. (2020), which uses the (empirical) daily estimates of CHIRPS precipitation and ensemble evaporation (2001-2012) (Appendix A1 and Sect. 2.3.2) (Fig. 1a). Since we are only interested in forest transitions, the ecosystems classified as savanna under the current climate are excluded from this analysis.

Next, for classifying ecosystems under future climate scenarios (Fig. 1b), we follow the same mass-balance approach (Appendix A1). However, since precipitation and evaporation estimates from ESMs do not align with empirical estimates (Baker et al., 2021; McFarlane, 2011), we employ a bias-correction method. Specifically, we use a histogram equivalence method (Piani et al., 2010) to adjust empirical $S_r$ thresholds to comparable CMIP6 $S_r$ thresholds for various ESMs (Table S1). This involves, first, calculating $S_r$ using CMIP6-historical precipitation and evaporation estimates between 2000-2014 (Appendix A1 and Fig. S8). We then determine percentile-equivalent $S_r$ thresholds for each of the thirty-three CMIP6-ESMs under the current climate. For example, if an empirical $S_r$ of 100 mm corresponds to the 10th percentile ($n = 20\%$ of total pixels), we find the 10th percentile in the CMIP6-historically $S_r$, which may be higher or lower than 100 mm for each ESM (Fig. 1

and Table S1). These percentile-equivalent $S_r$ thresholds are then used to classify ecosystems both under current
(CMIP6-hsitorical; 2000-2014) and future climate (CMIP6-SSPs; 2086-2100) (Fig. 1b).

Ultimately, we evaluate potential transitions by comparing ecosystems classified under current climate
conditions (*this excludes savanna*) with those under future climate conditions (*this includes savanna*) (Sect.
2.3.2). These transitions are divided into three distinct categories (Fig. 1c and Fig. A2):

i. **Forest-savanna transition:** This refers to current climate forest ecosystems that risk transitioning to a
255 savanna under future climate change. To classify savanna under future climate conditions, we assume
the ecosystem is in equilibrium with the projected climate (see detailed steps in Appendix A3).

ii. **Transition to a more water-limited state:** This includes ecosystems that shift to a higher water-limited
state in the future. For example, if a forest currently classified as lowly water-limited transitions to either
a moderately or highly water-limited state in the future, it would fall under this category.

iii. **Reversion to a less water-limited state:** This includes ecosystems that shift to a lower water-limited
state in the future.

To aggregate the results from all ESMs, grid cells with > 50% convergence are referred to as 'moderate-
high model agreement', 20-50% as 'moderate model agreement' and ≤ 20% as 'low model agreement'. In the
Results section, we primarily discuss estimates from scenarios >20% and >50% model convergence. While a
threshold of >20% may seem low given the total number of ESMs analysed, it is important to recognise the
variable and often limited capabilities of these ESMs, particularly in simulating biophysical interaction and
emerging properties due to our limited understanding of the Earth system (Lenton et al., 2019; Stevens and Bony,
2013). Opting for a majority-based consensus in ESMs could overlook critical tipping risks identified by a
minority of models, which might provide insights as valid as those from more widely agreeing models (Arora et
al., 2023; Reyer et al., 2015).

**2.4    Sensitivity analyses**
Our methodology operates under two key assumptions: (i) the empirically derived $S_r$ thresholds remain valid in
the future, and (ii) the hydroclimatic estimates projected by ESMs accurately represent the actual climate, even
though these models have prescribed land cover (Hurtt et al., 2020). To address the uncertainties related to the
first assumption, we conduct four sensitivity analyses to assess the robustness of our analysis: (a) assuming that
the regions exceeding the 99[th] percentile $S_r$ are prone to a forest-savanna transition, as high $S_r$ investment could
be unrealistic from the perspective of plants under future climate change, (b) evaluating forest transitions using
three different evaporation datasets, (c) assessing forest transitions under 10- and 40-year drought return periods,
and (d) adjusting the forest-savanna transition thresholds.

Regarding the second assumption, we explicitly apply this methodology across a wide range of available
ESMs under four SSP scenarios to identify consistencies and discrepancies in the results. Additionally, the

discrepancies between the prescribed land use and the forest transitions derived from our methodology, as well

as the implications of these assumptions, are detailed in the Discussion section.

## 3    Results

We find that under future climate conditions (2086-2100), considering >50% models' agreement, about one-

fourth of the forests in both South America and Africa are projected to transition (Fig. 2b-g). With >20% models'

agreement, these transitions are projected to occur for about three-fourths of the forests for both continents.

Considering a lower threshold for models' agreement causes double or triple counting of some transitions (Fig.

2b-g). To minimise this in further analyses, we only consider >50% models' agreement for forests that transition

to a more and less water-limited state. Furthermore, because (abrupt) forest-savanna transitions are under-

represented in ESMs (Drijfhout et al., 2015; Lenton, 2011; Maslin and Austin, 2012; Valdes, 2011), we consider

>20% models' agreement for them. Considering this, we not only reduce the overlap to <0.4% of the total forest

area (Fig. S9), but we also maximise highlighting forest-savanna transition risk for both continents.

We find that the risk of forest-savanna transitions mainly occurs in the Guiana Shield of South America,

and the southern and south-eastern regions of Africa (Fig. 3). Compared to Africa, forest-savanna transitions are

more prominent in South America under warmer climates (i.e., higher SSPs; Fig. 2b and 3). Our analysis reveals

that the extent of forest-savanna transitions in South America decreases from almost $1.32 \times 10^6$ km$^2$ (16.3% of

total forest area in South America) under the highest emission scenario to $0.04 \times 10^6$ km$^2$ (0.5%) under the lowest

emission scenario (Fig. 2b). Interestingly, for Africa, the extent of forest-savanna transition did not change much

for different SSPs, i.e., (median) $0.25 \times 10^6$ km$^2$ with a maximum deviation of $\pm0.11 \times 10^6$ km$^2$ (minimum and

maximum extent of transition between 3-6.6% of total forest area in Africa) (Fig. 2c).

When comparing the changes in forest-savanna transition risk areas relative to their immediate lower

warming scenarios, we find considerable increases for South America. The highest relative growth of

approximately 5.75 times is observed between SSP1 and SSP2, with the forest area under risk increasing from

$0.04 \times 10^6$ km$^2$ to $0.23 \times 10^6$ km$^2$, respectively. It increases by 3.48 times from SSP2 to SSP3 ($0.23 \times 10^6$ km$^2$ to

$0.80 \times 10^6$ km$^2$), and by 1.65 times from SSP3 to SSP5 ($0.80 \times 10^6$ km$^2$ to $1.32 \times 10^6$ km$^2$). For Africa, however,

the increases are more modest: the risk grows by 1.29 times from SSP1 to SSP2 ($0.17 \times 10^6$ km$^2$ to $0.22 \times 10^6$

311    km$^2$), by 1.63 times from SSP2 to SSP3 ($0.22 \times 10^6$ km$^2$ to $0.36 \times 10^6$ km$^2$), and is observed to decrease by 0.72

312    times from SSP3 to SSP5 ($0.36 \times 10^6$ km$^2$ to $0.26 \times 10^6$ km$^2$).

By evaluating changes to their hydroclimate, we find that under warmer climates, forest-savanna transition

regions in both continents are projected to experience a decrease in precipitation. Furthermore, we observe an

increase in precipitation seasonality for South America, whereas Africa shows a decrease (Fig. S12). Here, an

increase in precipitation seasonality (seasonal variability in precipitation over the year) creates water-limited

conditions for the ecosystem. In contrast, a decrease in seasonality and precipitation in Africa corresponds to a

lower moisture availability altogether. Nevertheless, for both these continents, this transition seems to occur for

the previously highly water-limited forests under the current climate, followed by moderately, with the least

contribution from lowly water-limited forests (Fig. 3). This highlights the looming risk on highly water-limited forests to experience a forest-savanna transition under warmer climates.

Forests that transition to a 'more' water-limited state in South America are spatially aggregated towards the border between Brazil, Colombia, and Peru – covering a considerable portion of the Central Amazon (Fig. 3). Whereas for Africa, these forests exist in moderate to small patches towards the northern and southern extent of central Congo rainforests. We observe that these transitions account for most of the projected changes to forests' states across both continents (Fig. 2d,e), with the transition to just the 'highly water-limited forest' accounting for more than three-fourths of all such transitions (Fig. 3). We observe that South American forests gradually become increasingly water-limited under warmer climates, with maximum and minimum projected transition of $1.89 \times 10^6$ km$^2$ (23.4%) and $1.61 \times 10^6$ km$^2$ (19.9%) observed under the highest and lowest emission scenarios, respectively (Fig. 2d,e). Whereas for Africa, the change in the water-limited state of the forests under different SSP scenarios remains almost similar (i.e., median 1.14 ($\pm$0.06) $\times 10^6$ km$^2$; 19.6-22.2%). Analysis of their hydroclimatic changes reveals that water-limitation is induced by both a decrease in precipitation and an increase in seasonality in South America (Fig. S13). In contrast, water-limitation in Africa is driven solely by an increase in seasonality. We observe that these newly water-limited forests seem to have permeated to regions that were previously (under the current climate) dominated by lowly and moderately water-limited forests (Fig. 3). Here, this shift only signifies the changes to hydroclimatic conditions allowing forests to transition to a more water-limited state, rather than the changes to the floristic composition of terrestrial species from one location to another. Although such a shift under changing climate is not unlikely (Esquivel-Muelbert et al., 2019), they are not analysed in this study.

Forests that revert to a 'less' water-limited state in South America are primarily observed in the south-eastern Amazon, with small patches observed towards eastern Brazil and the western coast of Equatorial Guinea and Gabon (Fig. 3). For Africa, the reverted forests exist in patches in the northern and southern regions of the Congo rainforest. Furthermore, for South America, we observe a gradual decrease in these reversions with an increase in warming. Here, we observe the lowest reversion of $0.23 \times 10^6$ km$^2$ (2.8%) under the highest emission scenario and the highest reversion of $0.67 \times 10^6$ km$^2$ (8.4%) under the lowest emission scenario (Fig. 2f,g). For Africa, these trends remain almost similar under all SSPs (i.e., median 0.18 ($\pm$0.05) $\times 10^6$ km$^2$; 2.2-3.5%). Comparing these transitions with their hydroclimatic changes reveals an overall increase in precipitation (Fig. S14). Interestingly, we observe a much higher precipitation increase for South America under high-emission scenarios than those in lower-emission scenarios. However, we find that precipitation seasonality is also higher for these ecosystems under warmer climates (Fig. S14). This suggests that increased precipitation without changes to precipitation seasonality helps decrease the water-limitation of the ecosystem, compared to the ecosystems that experienced a simultaneous increase in both.

Our sensitivity analysis, detailed in Appendix B1, reveals a consistent pattern of forest transitions across various scenarios.

# 4 Discussion

## 4.1 Asynchronous resilience risks under future climate change

Our analysis reveals the spatial extent of potential ecosystem transitions in South America and Africa and their vulnerability to future climate change (Fig. 2 and 4). For South America, we find a clear indication of a decrease in forest resilience (i.e., an increase in water-limited forests) and an increase in forest-savanna transition risk under warmer climates (Fig. 2b,d,f). In contrast, these trends are not symmetric for Africa, where transition risk shows only slight variation across the different SSPs (Fig. 2c,e,g). Similar to the results of this study, previous studies on rainforest tipping have also suggested that exceeding 1.5-2°C will considerably increase the tipping risk (Flores et al., 2024; Jones et al., 2009; Parry et al., 2022), with the Guyana Shield in the Amazon being the most susceptible under future climate change (Cox et al., 2004; Staal et al., 2020) (Fig. 3 and Table S3). Previous studies also agree that, in contrast to the Amazon, the projected risk to Congo rainforests is not substantial (Higgins and Scheiter, 2012; Staal et al., 2020) (Fig. 2). Despite it being unclear to what extent the ESMs represent the correct carbon-water dynamics (Koch et al., 2021), our results show a further divergence between Amazon's and Congo's responses to different SSPs (Fig. 2 and Fig. S12-S14). This could either be caused simply by a different response to changes in precipitation patterns over the respective regions (Kooperman et al., 2018; Li et al., 2022) or a different response to increased $CO_2$ levels in the atmosphere (Brienen et al., 2015; Hubau et al., 2020; Trumbore et al., 2015).

Previous empirical studies have linked these divergent responses to evolutionary and biogeographical differences between the ecosystems, which resulted in distinct species pools that uniquely influence each ecosystem's adaptability and response to climate change (Fleischer et al., 2019; Hahm et al., 2019; Hubau et al., 2020; Slik et al., 2018). These studies found that forest ecosystems in the Amazon tend to be more dynamic – grow faster due to high $CO_2$ levels in the atmosphere – than those in the Congo rainforests. However, these fast-growing trees also die young due to them investing substantially less in their adaptive strategies against perturbations than (less dynamic) old-growth forests (Brienen et al., 2015; Körner, 2017; Rammig, 2020). This makes the Amazon rainforest especially sensitive to $CO_2$ emissions pathways, as the positive influence of $CO_2$ fertilisation-induced growth is counteracted by the negative impact of warming and droughts, thereby exacerbating the risk of forest mortality under high emission scenarios (Brienen et al., 2015; Hubau et al., 2020; Yang et al., 2018). In this case, the projected changes to the future hydroclimate could be an artefact of decreased transpiration and precipitation due to forest mortality, rendering the rainforests vulnerable to tipping. In contrast, terrestrial species in Congo rainforests appear more resilient, having adapted to severe droughts during glacial periods, which makes them better equipped to handle episodic water-induced perturbations than Amazon (Cole et al., 2014).

Nevertheless, with compounding influence from land-use and climate-induced hydroclimatic changes (Davidson et al., 2012), these rainforests risk tipping to a savanna state. Our results highlight that by keeping the mean global surface temperature below 1.5-2ºC warming (which in this case is equivalent to SSP1-2.6 relative

392 to the pre-industrial), we minimise forest-savanna transition risk and maximise recovery – thereby improving the
393 resilience of rainforest ecosystems (Fig. 2, 3 and 4).

395 **4.2 Changes in atmospheric moisture flow drives forest-savanna transition**

396 Among all transitions, the most noticeable and catastrophic (since it is difficult to revert) is the forest-savanna
397 transition projected in the Amazon's Guiana Shield of South America, and over the southern and south-eastern
398 parts of Africa (Fig. 3 and 4). These transitions are associated with the shifting of the inter-tropical convergence
399 zone (ITCZ) (Mamalakis et al., 2021), which decreases precipitation and increases precipitation seasonality over
400 the continents. For South America, the creation of these low-pressure bands allows the trade winds to bring in
401 considerable moisture from the equatorial Atlantic Ocean over to Amazon by passing through the Guiana Shield
402 and ultimately carrying it across the La Plata Basin via the South American low-level jet (Bovolo et al., 2018;
403 van der Ent et al., 2010; Zemp et al., 2014). Similarly, for Africa, south-eastern trade winds bring moisture from
404 the Indian Ocean over the centre of the African continent (Mamalakis et al., 2021).

405   Under a warmer climate, sea surface temperature over the equatorial Atlantic and the northern Indian
406 Ocean is projected to increase (Pascale et al., 2019; Zilli et al., 2019), leading to a southward shift in ITCZ over
407 the eastern Pacific and Atlantic Oceans, and northward over east Africa and the Indian Ocean (Mamalakis et al.,
408 2021; Xie et al., 2010). Previous studies also acknowledge that the intense surface warming over the Sahara under
409 future climate can also attract ITCZ northwards in Africa (Cook and Vizy, 2012; Dunning et al., 2018; Mamalakis
410 et al., 2021). These climate change-induced shifts in ITCZ can potentially both mitigate and exacerbate the effects
411 of (accumulated) water-deficit on the forest ecosystem, especially critical for highly water-limited forests, even
412 without considering the changes to atmospheric moisture flow caused by localised deforestation (Leite-Filho et
413 al., 2021; Schumacher et al., 2022; Staal et al., 2018; Wunderling et al., 2022). This underscores the importance
414 of including changes in atmospheric circulation in studies that analyse the impact of future climate on the
415 resilience of forest ecosystems (Staal et al., 2020; Zemp et al., 2017).

418 **4.3 Discrepancy between prescribed future land use and projected transitions**

419 The land-use information in CMIP6-ESMs is not biophysically simulated, but prescribed based on simulations
420 from Integrated Assessment Models (IAMs) for each SSP scenario (Hurtt et al., 2020). Therefore, it is valuable
421 to examine whether these prescribed land-use scenarios agree or conflict with the changes projected (assuming
422 equilibrium between hydroclimate and the ecosystem) by our $S_r$-based ecosystem transitions (Fig. 5 and Fig. S15-
423 S17).

424   The most noticeable discrepancies are observed in South America, where the extent of forest-savanna
425 transitions is underestimated in prescribed land-use scenarios compared to those projected in this study (i.e.,
426 prescribed land-use predicts forests in the region whose hydroclimate can't support forest; Fig. 4 and 5a).
427 Additionally, in South America, our analysis highlights the potential of some forests reverting to a 'less water-

limited state' in places where the prescribed land use in the ESMs suggests non-forest landscapes (Fig. 4 and 5c). These discrepancies arise because the prescribed land use in CMIP6-ESMs does not shift in response to hydroclimatic changes. Despite our approach assuming equilibrium and overlooking the temporal dynamics of transitions, based on broad climate change patterns (Sect 4.2), we believe it more accurately represents the ecohydrological state of the ecosystems.

However, these prescribed land uses can introduce errors in subsequent biophysical processes simulated in ESMs (Ma et al., 2020), affecting the accuracy of projected transitions. For example, prescribing a region as a forest that would be grassland in the future will lead to the extraction of deeper subsoil moisture in ESMs, which (actual) grasslands do not have the capacity to access (Ahlström et al., 2017; Yu et al., 2022). This will result in an overestimation of the ecosystem's evaporation, potentially altering precipitation patterns downwind and leading to inaccurate water budget assessments for these ecosystems. Consequently, causing erroneous projections of the ecosystem state. These discrepancies underscore the urgent need for enhancements in the land surface components of ESMs, enabling dynamic simulations of vegetation-climate feedbacks. Such improvements would provide a more accurate representation of the ecohydrology of terrestrial ecosystems and their response to changing climate conditions.

## 4.4    Limitations

This study assumes that the $S_r$-derived thresholds used to classify terrestrial ecosystems under current climate conditions remain valid under future climate change. However, forests themselves are dynamically adapting their structure and functions in response to climate change, altering their critical thresholds (Doughty et al., 2023). Thus, assuming a static critical threshold may lead to inaccuracies in estimating forests' resilience to future climate change. For instance, under the $CO_2$ fertilisation effect, forests may become more water-use efficient (i.e., transpire less and therefore need for a lower $S_r$) (Xue et al., 2015), potentially delaying their tipping under warming scenarios compared to those projected in this study. Conversely, factors such as nutrient limitation (Condit et al., 2013) or extensive human influence (van Nes et al., 2016) in the ecosystem might lead to an earlier tipping than anticipated.

However, the uncertainty surrounding the effect of $CO_2$ fertilisation, nutrient limitation, and human influence on vegetation remain significant research frontiers for enhancing our understanding of rainforest tipping under future climate change (Fleischer et al., 2019; Hofhansl et al., 2016). Additionally, factors such as precipitation variability, species composition, soil properties, and topography can contribute to varied local-scale forest responses to future climate change (Staal et al., 2020). It should also be noted that though these uncertainties may hinder our understanding of local-scale forest resilience, the influence of future hydroclimatic changes on forests still constitutes major prediction uncertainties. Therefore, in this study, regardless of how these influences are parametrised or simulated in each ESM, we assume that hydroclimatic estimates projected by the ESMs represent the actual climate.

Of course, this assumption opens us and other studies projecting forest conditions to future climate change to certain limitations. Our ability to project forest-savanna transitions (or any transition) relies on the model's capacity to simulate complex feedbacks. Some models capture complex vegetation-atmosphere interaction, simulating local and regional scale feedbacks across time (Ferreira et al., 2011; Jach et al., 2020); others rely on simpler parametrisation (Nof, 2008) (e.g., parametrisation of $CO_2$ fertilisation; Koch et al., 2021). However, caution should be taken to not overgeneralise the functioning of tropical forests just from the analysis presented in this study, and also realise the current potential of ESMs to simulate them (Staal et al., 2020). We believe that by considering simulations from multiple ESMs under different SSP scenarios, not only do we highlight the agreements and conflicts between potential transitions; but also allow future studies to disentangle vegetation-climate feedbacks and improve the modelling of local-scale interactions (e.g., vegetation's water-uptake profile, species response to $CO_2$ fertilisation) in the ESMs.

## 5 Conclusions

Classifying terrestrial ecosystems based on empirical and CMIP6 ESMs-derived $S_r$ allowed us to assess the future transitions in the rainforest ecosystems. Our findings indicate that the climate projected under the lowest emission scenarios significantly reduces the risk of rainforest tipping and maximises reversion to a less water-limited state, while the climate projected under the high emission scenarios has the opposite effect on the forest ecosystem. Specifically, in the Amazon rainforest, the risk of forest-to-savanna transition increases considerably with incremental increases in warming. Conversely, in the Congo, the variation in transition risk across different emission scenarios is relatively minor.

Notably, our analysis suggests a very limited tipping risk that is 'unavoidable' (i.e., regions prone to a forest-savanna transition in all scenarios), and the vast majority of potential transition risks can still be avoided by steering towards a less severe climate scenario, thereby underscoring the critical window of opportunity. Moreover, regions projected to revert to a less water-limited state could potentially become more amenable to restoration and responsive to deforestation prevention efforts. This study highlights the importance of restricting global temperature change below 1.5-2$^{\circ}$C warming relative to the pre-industrial levels to prevent forest tipping risks and provide the best conditions for effective ecosystem stewardship.

## Appendix A: Methodology

### A1. Root zone storage capacity calculation

Our method to calculate $S_r$ is adopted from Singh et al. (2020). For estimating $S_r$, we first obtained the water deficit ($D_t$) at daily time step from the daily estimates of precipitation ($P_t$) and evaporation ($E_t$) (Fig. A1) using:

$$D_t = E_t - P_t \tag{A1}$$

Here, $t$ denotes the day count since the start of the simulation, with simulation for each grid starting in the month with maximum precipitation. Second, we calculated the accumulated water deficit integrated at each one-day timestep for one year using:

$$D_{a(t+1)} = \max\{0, D_{a(t)} + D_{t+1}\} \tag{A2}$$

Where $D_{a(t+1)}$ is the accumulated water deficit at each time step (Fig. A1). Here, an increase in the accumulated water deficit will occur when $E_t > P_t$, and a decrease when $E_t < P_t$. However, since this algorithm estimates a running estimate of root zone storage reservoir size, we use a maximum function to calculate the accumulated water deficit, which by definition can never be below zero. Not allowing $D_{a(t+1)}$ to be negative also means that excess moisture from precipitation will either contribute to deep drainage or runoff. Lastly, the maximum accumulated annual water deficit ($D_{a,y}$) will represent the maximum storage required by the vegetation to respond to the critical dry periods (Fig. A1).

$$D_{a,y} = \max\{D_{a(t+1)}\} \quad t = 1 : n-1 \tag{A3}$$

This simulation runs for a whole year, with $n$ denoting the number of days in year $y$.

Different terrestrial ecosystems (e.g., forest, savanna and grassland) adapt to different drought return periods (de Boer-Euser et al., 2016; Gao et al., 2014; Wang-Erlandsson et al., 2016). For instance, grasslands and savannas adapt to shorter drought return periods (i.e., <10 years and 10-20 years, respectively). In contrast, forests adapt to long drought return periods (>40 years) (Wang-Erlandsson et al., 2016). For this study, we use a uniform 20-year drought return period (following Bouaziz et al., 2020; Nijzink et al., 2016) to avoid any artificially introduced transitions between different ecosystems. Thus, this 20-year drought return period $S_r$ refers to the maximum amount of root zone moisture accessible to vegetation for transpiration during the largest accumulated annual water deficit expected every twenty years under static climate conditions. We analyse this using the Gumbel extreme value distribution (Gumbel, 1958) and apply it to normalise all $\overline{D_{a,y}}$. The Gumbel distribution ($F(x)$) is given by:

$$F(x) = \exp\left[-\exp\left[-\frac{(x-\mu)}{\alpha}\right]\right] \tag{A4}$$

Where $\mu$ and $\alpha$ are the location and scale parameters, respectively. We calculate this using the python package 'skextremes'(skextremes Documentation):

$$S_r = \overline{D_{a,y}} + K \times \sigma_{n-1} \tag{A5}$$

Where $K$ is the frequency factor given by:

$$K = \frac{y_t - y_n}{S_n} \tag{A6}$$

And $y_t$ is the reduced variate given by:

$$y_t = -\left[\ln\left[\ln\left(\frac{T}{T-1}\right)\right]\right] \tag{A7}$$

Where $T$ is the drought return period (i.e., 20 years used in this study), $\overline{D_{a,y}}$ is the mean annual

accumulated deficit for the years 2001-2012, $\sigma_{n-1}$ is the standard deviation of the sample. Also, $y_n$ is the reduced

mean and $S_n$ is the reduced standard deviation, which for $n = 11$ years (since we are calculating $S_r$ in a hydrological

530 year – simulation starts mid-year – we therefore lose one year) is equal to 0.4996 and 0.9676, respectively

(Gumbel, 1958).

Since the CMIP6 (-historical and -SSP estimates, the timeframe considered are 2000-2014 and 2086-

2100, respectively) doesn't have daily estimates of evaporation and precipitation for all Earth System Models

(ESMs), we directly use the monthly estimates of precipitation and evaporation to modify Eq. (A1) as:

$$D_t = E_{t(monthly)} - P_{t(monthly)} \tag{A8}$$

Here, $t(monthly)$ denotes the month count since the start of the simulation. The rest of the steps (Eq. A2-

A7) remain the same for CMIP6 datasets. For CMIP6 runs, $y_n$ and $S_n$ in Eq. (6) are calculated for $n = 14$ years

(Eq. A7) equal to 0.5100 and 1.0095, respectively. The $S_r$ estimates derived from daily and monthly empirical

estimates (from Eq. A1 and A8) are compared in Fig. S8 to evaluate uncertainty.

## A2. Abiotic and biotic factors influence soil moisture availability

In this study, $S_r$ quantifies the hydrological buffer necessary for an ecosystem to maintain its structure and

functions, reflecting the amount of root zone soil moisture available to vegetation for transpiration. Our mass-

balance-based $S_r$ methodology, while not directly distinguishing between the biotic and abiotic influences on soil

moisture and root characteristics, does incorporate their critical role in shaping the ecohydrology of the ecosystem

under climate change. By utilising empirical precipitation and evaporation data, our approach theoretically

captures the combined impact of these biotic and abiotic factors on the actual hydrological regime (including soil

moisture) of the ecosystem (Sect. 2.3.2).

We acknowledge that abiotic factors such as soil texture, structure, and depth profoundly affect soil

water-holding capacity (Fayos, 1997). For instance, field studies suggest that clay and organic-rich soils exhibit

superior water retention capabilities due to their fine textures and high surface areas, crucial to vegetation for

moisture uptake during extended dry periods (Bronick and Lal, 2005; Fayos, 1997). Additionally, the depth and

porosity of soil also dictate its ability to absorb and store water in the soil, with deeper, less compacted soils

providing a higher buffer against drought by allowing greater water infiltration (Indoria et al., 2020; Smith et al.,

2001). By altering temperature and precipitation patterns, climate change can modify these abiotic soil properties,

potentially leading to a loss in soil water retention capacity through erosion and compaction (Dexter, 2004).

Moreover, biotic factors, including plant-root dynamics and microbial activity, also play essential roles in shaping the ecosystem (Brunner et al., 2015; Sveen et al., 2024). Deep and extensive root systems not only directly improve access to deeper soil moisture, but also physically modify the soil to enhance its permeability and storage (Canadell et al., 1996; Jackson et al., 1996). Additionally, microbial processes contribute by breaking down organic matter, thereby improving the soil's structural integrity and ability to retain water (Dittert et al., 2006). These biotic interactions, coupled with changing abiotic factors under climate change, underscore the complex dynamics that govern soil moisture availability and ecosystem resilience. However, this study does not consider the direct impact of future climate change on biotic and abiotic factors, nor their influence on ecosystems, beyond changes to $S_r$.

**A3. Using precipitation to discern savanna from forests under future climate change**

Under future climate change, some ecosystems will remain forest, while others may transition to savanna. In our $S_r$-based framework, without information about above-ground forest structure, it is difficult to discern whether an ecosystem is a forest or savanna just with $S_r$ (for instance, an ecosystem with $S_r$ of 200 mm can either be a moderately water-limited forest or savanna; Sect. 2.3.2). Differentiating these ecosystems is easier under the current climate, where we have several remote sensing products capturing vegetation structure (e.g., tree cover density, tree height, floristic patterns) (Aleman et al., 2020; Hirota et al., 2011; Xu et al., 2016). However, under future climate, we must find a proxy, since land-use information in ESMs is prescribed (i.e., not biophysically simulated) (Ma et al., 2020).

To address this, previous studies have either relied on vegetation structure proxies provided by ESMs (e.g., net primary productivity) (Boulton et al., 2013; Jones et al., 2009) or assumed that terrestrial ecosystems are in equilibrium with their climate (Staal et al., 2020) (see Supplement). In this study, we adopted the latter approach and utilised climate variables, specifically (bias-corrected) mean annual precipitation and the precipitation seasonality index, as proxies to make this distinction (Fig. S4). The climate conditions (or range) necessary for forest ecosystems to sustain themselves are determined by comparing empirical estimates of mean annual precipitation and precipitation seasonality index with $S_r$. These estimates are then bias-corrected (following the same methods described in Sect. 2.3.3) before applying them to future climate scenarios. This (revised) classification of terrestrial ecosystems is then used to assess forest transitions under future climate change scenarios.

# Appendix B:  Results

**B1. Sensitivity analysis reveals robust performance of the framework**

Sensitivity analysis reveals that by setting an extreme $S_r$ threshold – signifying a forest-savanna transition for ecosystems that cannot maintain their above-ground structure at high $S_r$ – we observe some shifts near the already projected risk regions and coastal areas (Fig. 3 and Fig. S18). However, the transition risk identified in the coastal

regions may be an artefact of interpolating hydroclimate estimates to higher resolution. Additionally, since evaporation is more prevalent over oceans than land, this could result in high $S_r$ values, thereby projecting an elevated tipping risk in these coastal areas.

We also discover that variations in the evaporation datasets and return periods used for calculating $S_r$ have minimal effect on forest transitions (Fig. S19 and S20). Although the forest classification thresholds may shift with different evaporation products under current climate conditions (Singh et al., 2020), our histogram equivalence method ensures that forest classifications under future climates adjust accordingly, resulting in only minor alterations to the final outcome (Fig. 1b and Fig. S19). Furthermore, while $S_r$ values tend to increase with increase with shorter return periods, the impact of these changes becomes less significant with longer return periods (Wang-Erlandsson et al., 2016), leading to minor variations in the end results (Fig. S20).

Moreover, lowering the forest-savanna transition thresholds can reduce the risk of forest-savanna transition since it expands the associated range of climate conditions (i.e., mean annual precipitation and seasonality) necessary for forests to sustain their structure and functions (Fig. S21). Conversely, increasing the forest-savanna transition threshold leads to an opposite trend, where the risk of transition increases (Fig. S22). Despite these sensitivity analyses, the variation in transition magnitudes is minor, and the trends across different SSP scenarios for both continents remain consistent (Fig. 2 and Fig. S18-S22). Therefore, the conclusions drawn from this study remain robust, even with variations in factors that could potentially affect forest transitions.

## Code availability

The python-language scripts used for the analyses presented in this study are available from GitHub: https://github.com/chandrakant6492/Future-forest-transitions-CMIP6. The python-language code for calculating (empirical) root zone storage capacity is available from GitHub: https://github.com/chandrakant6492/Drought-coping-strategy.

## Data availability

All the data generated during this study is made publicly available at Zenodo: https://zenodo.org/record/7706640. Other datasets that support the findings of this study are publicly available at: (CMIP6; citations referred to in Table S2) https://aims2.llnl.gov/, (Root zone storage capacity; empirical) https://github.com/chandrakant6492/Drought-coping-strategy, (P-CHIRPS) https://data.chc.ucsb.edu/products/CHIRPS-2.0/, (E-BESS) ftp://147.46.64.183/, (E-FLUXCOM) ftp.bgc-jena.mpg.de, (E-PML) https://data.csiro.au/collections/#collection/CIcsiro:17375v2, (E-ERA5) https://cds.climate.copernicus.eu/cdsapp#!/dataset/reanalysis-era5-single-levels, (Globcover) http://due.esrin.esa.int/page_globcover.php. Potential transitions for each ESM based on the comparison between empirical (2001-2012) and SSP (2086-2100) scenarios are presented in the Supplement.

## Author contribution

All authors contributed to the conceptualisation of this research. CS performed the analyses and wrote the initial draft. All authors contributed to the discussion and revisions, leading to the final version of the manuscript.

## Competing interests

The authors declare that they have no conflict of interest.

## Acknowledgements

C.S., I.F. and L.W.-E. acknowledge funding support from the European Research Council (ERC) project 'Earth Resilience in the Anthropocene', project number ERC-2016-ADG-743080. L.W.-E. also acknowledges funding support from the Swedish Research Council for Sustainable Development (FORMAS), project number 2019-01220 and the IKEA Foundation. R.v.d.E. acknowledges funding support from the Netherlands Organisation for Scientific Research (NWO), project number 016.Veni.181.015. The authors also acknowledge the computational support provided by Microsoft Planetary Computer (https://planetarycomputer.microsoft.com) for performing the analyses.

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

## Figures

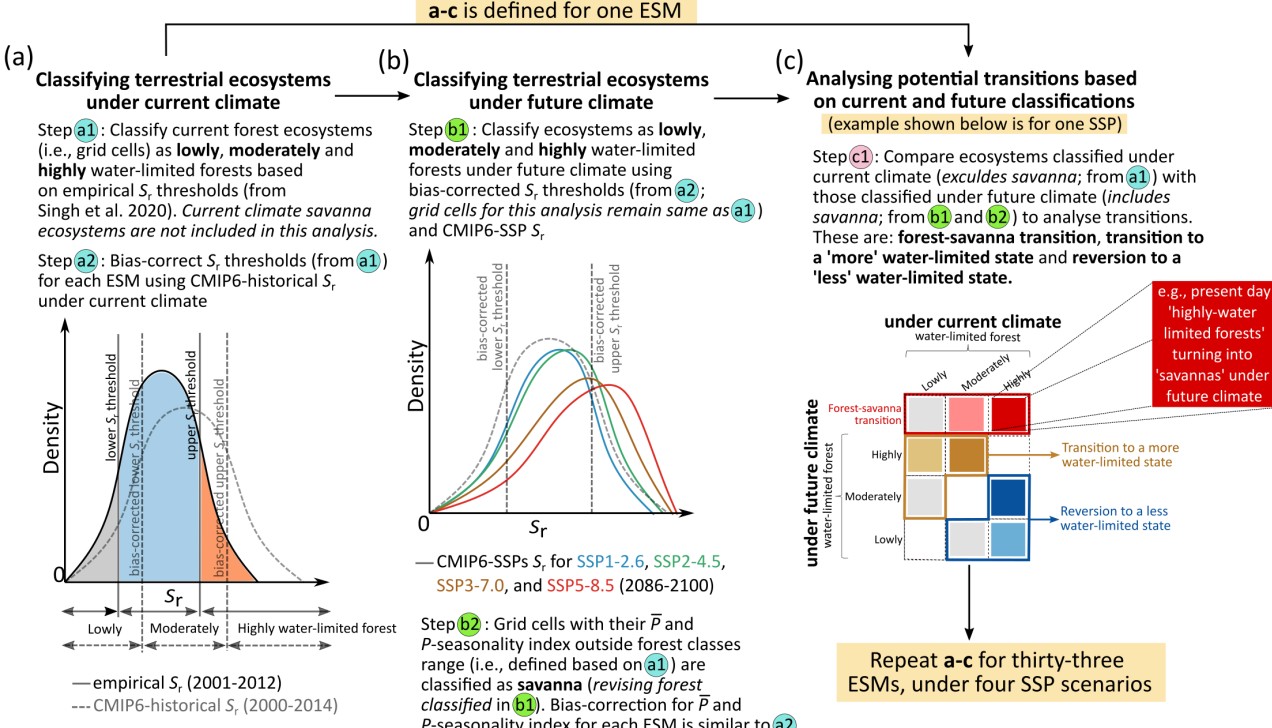

**Figure 1: Methodological framework for analysing the potential transitions in tropical terrestrial ecosystems using empirical and CMIP6-Earth System Models (ESMs) hydroclimate estimates. (a)** We use root zone storage capacity ($S_r$)-based classification thresholds (obtained from Singh et al., 2020) – calculated using empirical precipitation ($P$) and evaporation ($E$) estimates (Fig. S1; see Methodology section and Appendix A1) – to classify terrestrial ecosystems under the current climate. Savanna ecosystems under the current climate are excluded from this analysis. We bias-correct these $S_r$ thresholds for all ESMs using the histogram equivalence method (Piani et al., 2010) (Table S1). **(b)** We then use these bias-corrected $S_r$ thresholds to classify ecosystems under future climate conditions (Fig. S2 and S3). Furthermore, we use mean annual precipitation ($\bar{P}$) and $P$-seasonality index range ($S_r$-based forest classes from a) - as a proxy for ecosystem state - to revise our classification under future climate (Appendix A3 and Fig. S4). **(c)** We then analyse the potential transitions by comparing ecosystems classified under the current climate (analysed in a) with those classified under future climate (analysed in b) individually for all ESMs (Fig. S5 and S6). The transition analysis assumes that the hydroclimate and the ecosystem are in equilibrium, and does not account for the time required for transitions to occur. A detailed description is provided in the Methodology section. An exemplification of this methodological framework is shown in Fig. S7.

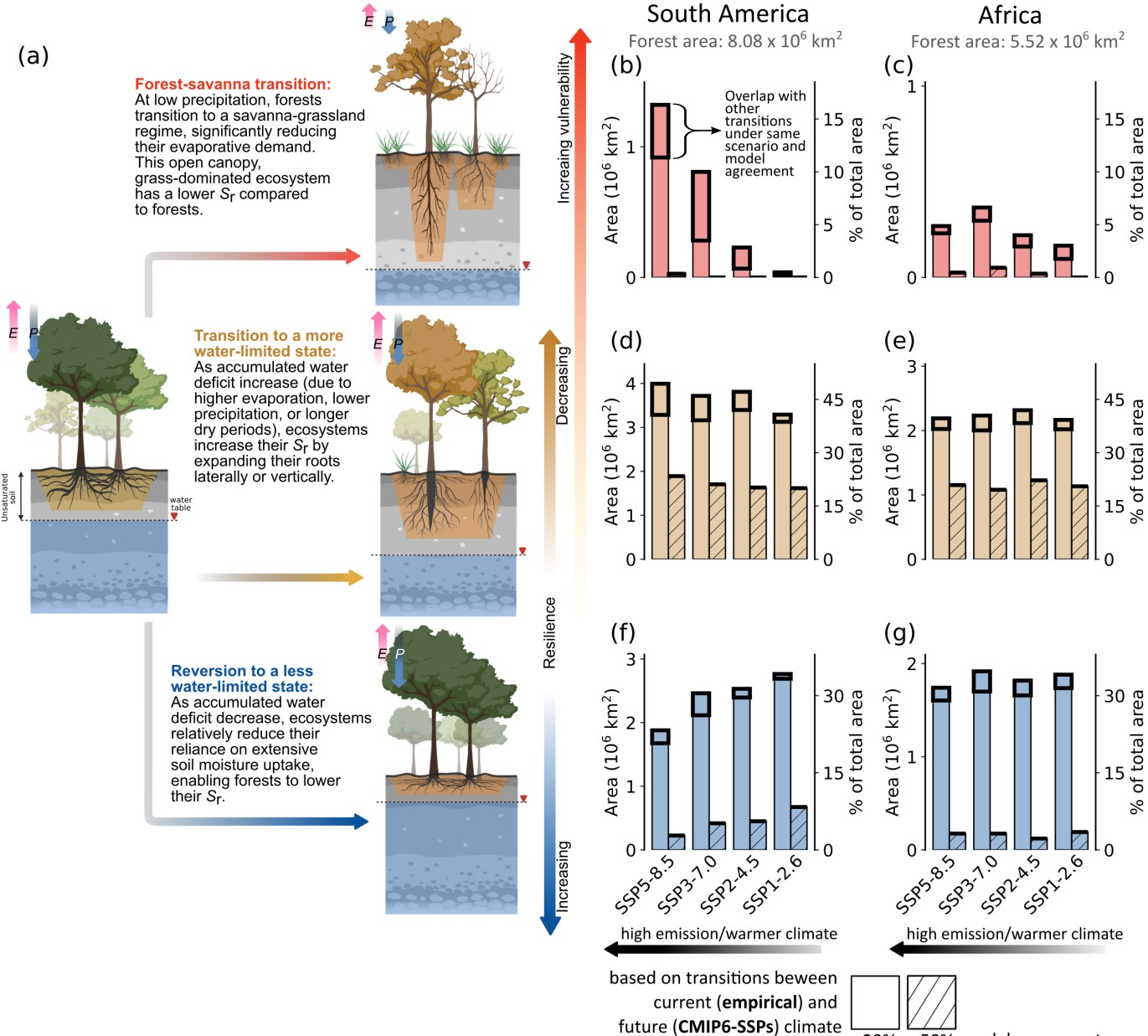

**Figure 2: Comparing the potential transitions under different SSP scenarios. (a)** The state of the ecosystem, both above- and below-ground, (post-transition) under future climate, quantifying **(b,c)** forest-savanna transition, **(d,e)** forests' that transition to a more water-limited state and **(f,g)** revert to a less water-limited state for South America and Africa (present forest area mentioned on the top of (b,c)), respectively. For the analysis above, transitions are calculated for grid cells with model agreement >20% (plain bar plot) and >50% (hatched bar plot). These quantifications show changes in the forest area based on ecosystem transitions under empirical-current (2001-2012) and future (2086-2100) climate conditions. For all these transitions, we assume that the hydroclimate and the ecosystem are in equilibrium. Analyses comparing ecosystem transitions based on CMIP6-historical (2000-2014) and future (2086-2100) climate conditions are shown in Fig. S10 and S11. For each transition, the total area of spatial overlap with other transitions under the same SSP scenario and model agreement is highlighted with thick black bars. The $P$ and $E$ arrows in (a) describe the relative magnitude of precipitation and evaporation fluxes. The illustration in (a) is adapted from Singh et al. (2020) and created with BioRender.com.

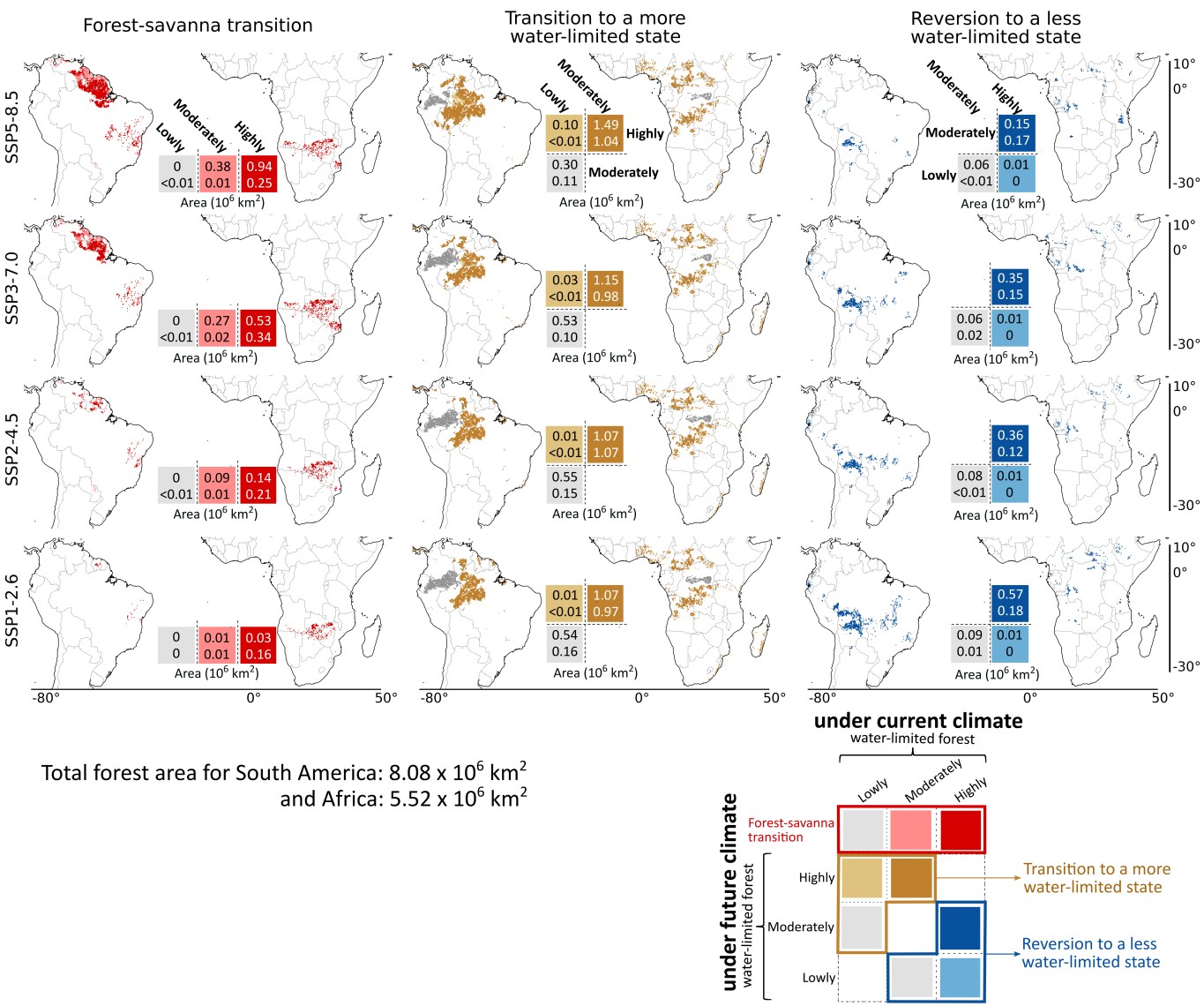

Total forest area for South America: $8.08 \times 10^6$ km$^2$
and Africa: $5.52 \times 10^6$ km$^2$

**Figure 3: Spatial extent of potential transitions with respect to their current state under different SSP scenarios.** We analysed transitions, explicitly focusing on forest-savanna transition, transition to a more water-limited state, and reversion to a less water-limited state, by comparing different ecosystem classes under current (empirical; 2001-2012) and future (SSPs; 2086-2100) climate conditions (as defined in Fig. 2). All transitions shown above are analysed for moderate-high (>50%) model agreement, except forest-savanna transition, for which moderate (>20%) model agreement is considered. Values overlaying the legends correspond to the total area of transition for South America (top values) and Africa (bottom values).

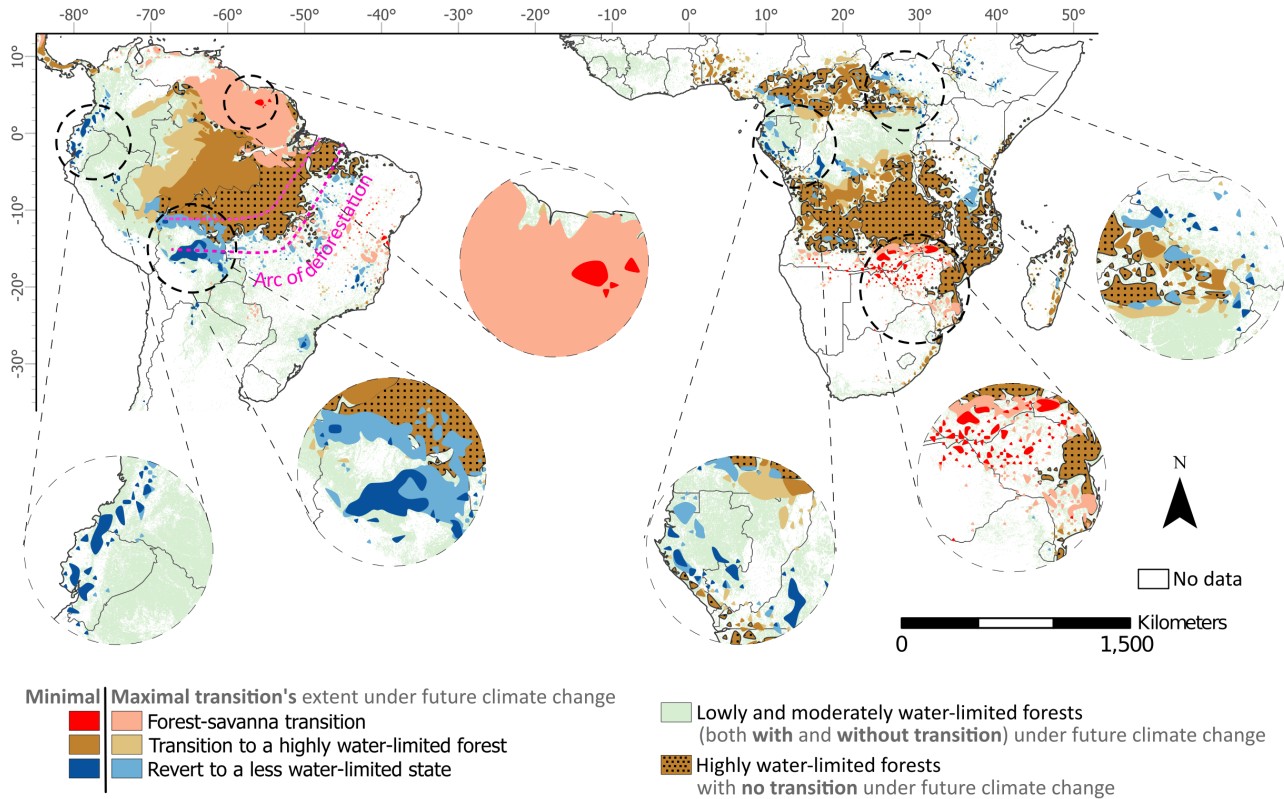

**Minimal | Maximal transition's** extent under future climate change
- ■ Forest-savanna transition
- ■ Transition to a highly water-limited forest
- ■ Revert to a less water-limited state

- Lowly and moderately water-limited forests (both **with** and **without transition**) under future climate change
- Highly water-limited forests with **no transition** under future climate change

**Figure 4: Minimal and maximal extent of potential ecosystem transitions under future climate change in**
**the entire study region over South America and Africa**. The three transition types are: forest-savanna
transition, from any class to highly water-limited forests, and to a less water-limited state (see definitions in Fig.
2 and 3). For better visualisation of these transitions, in this figure, we first converted all grid cells to shape,
merged them, and then smoothed them using the 'polynomial approximation with exponential kernel' function
(with a tolerance value of 1) in ArcGIS pro. The unsmoothed version of the transitions is shown in Fig. 3. The
minimal and maximal represent the minimum and maximum possible extent of transitions (as quantified in Fig.
3) based on changes between current (empirical; 2001-2012) and future (SSPs; 2086-2100) climate conditions
regardless of the SSP scenarios.

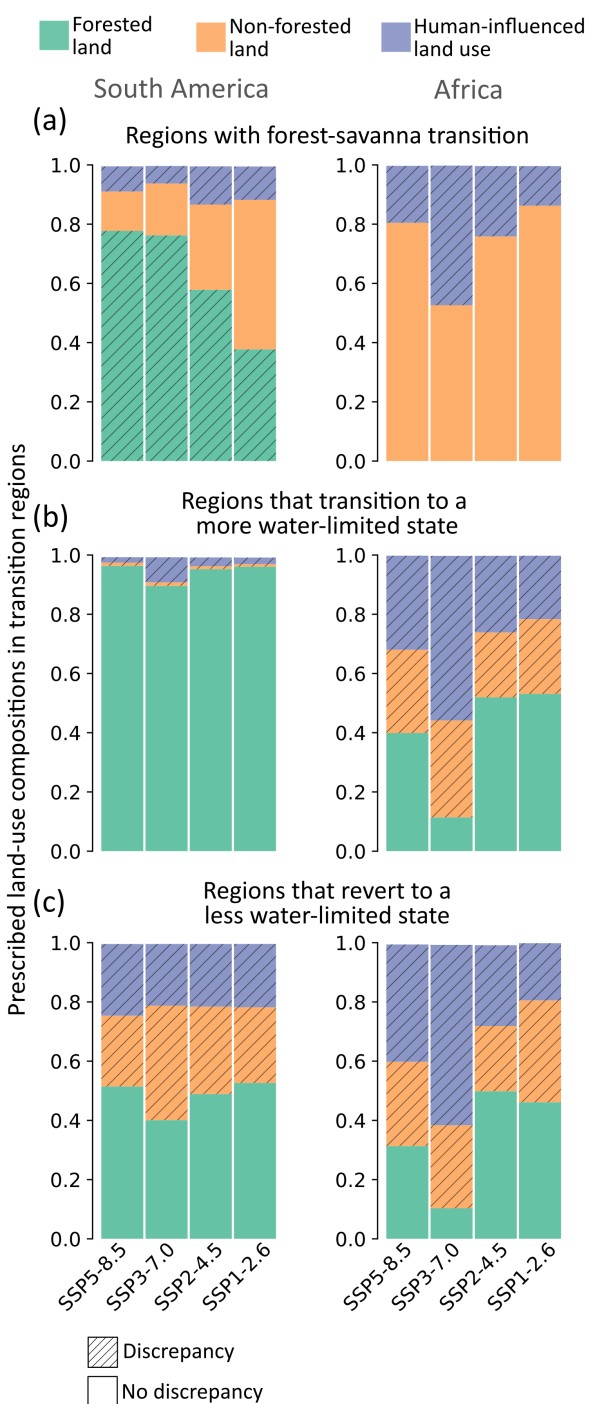

**Figure 5: Prescribed land-use composition for each transition region under different SSP scenarios (median 2086-2100), calculated as the ratio between the prescribed land use area and the projected transition area.** Regions where IAM prescribed land use are same as the projected transitions (from Fig. 3) are shown in plain colours (i.e., no discrepancy). Whereas regions where IAM-prescribed land use differs from projected transitions are hatched (i.e., discrepancy).

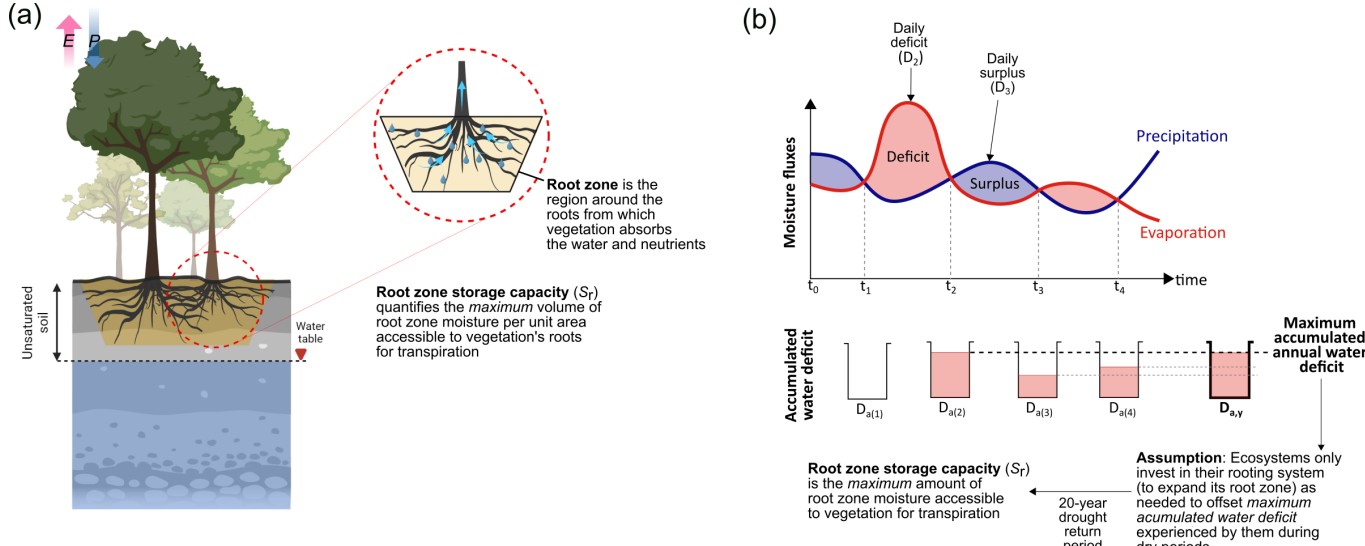

1172

**Figure A1:** The figure illustrates the root zone storage capacity ($S_r$) of the ecosystem. (a) We show the difference between the ecosystem's root zone and how that constitutes its $S_r$. (b) Conceptual illustration of how the ecosystem's precipitation and evaporation fluxes constitute the maximum accumulated annual water deficit ($D_{a,y}$) and $S_r$. The figure is adopted from Singh (2023) and Wang-Erlandsson et al. (2016).

1177

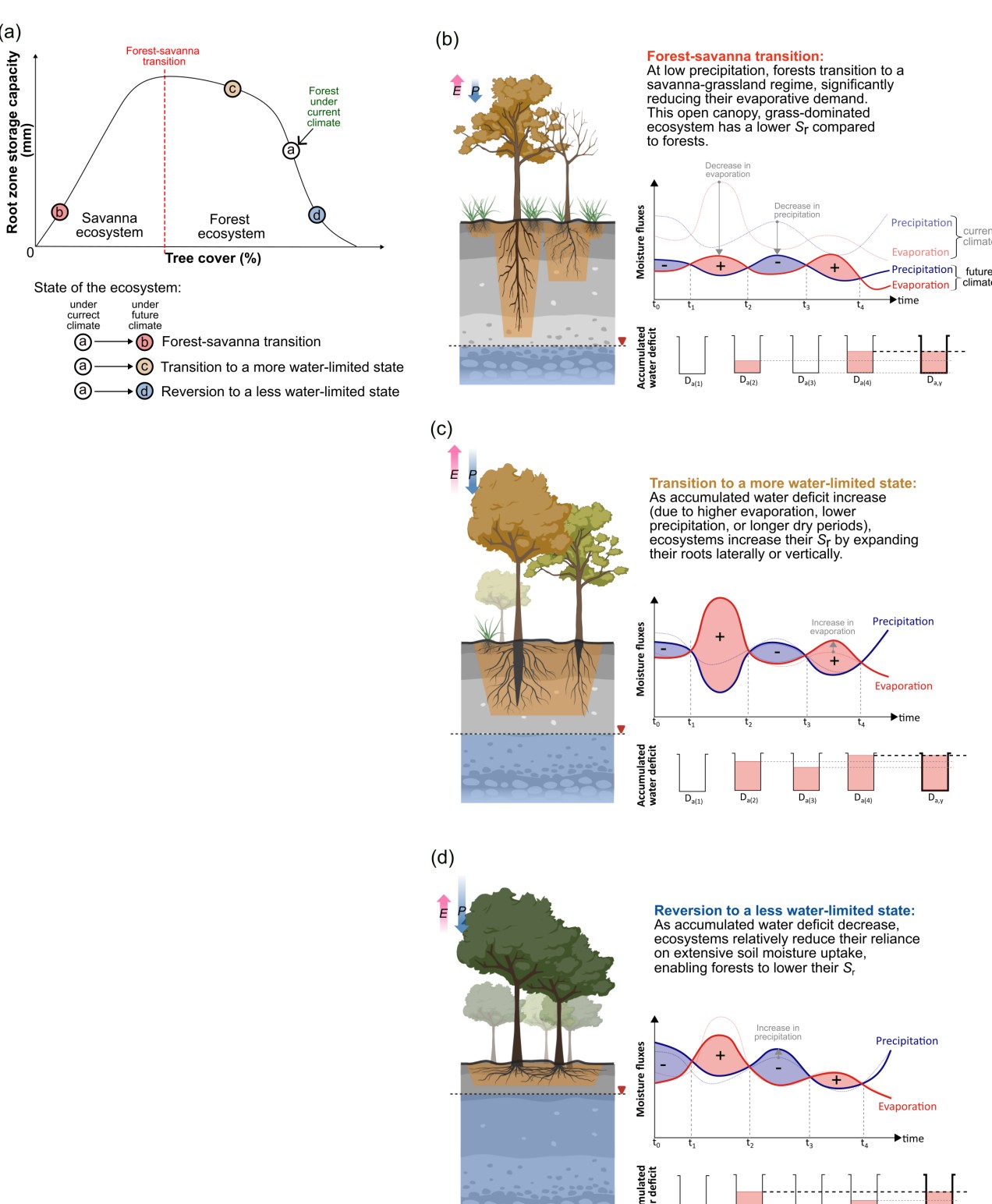

**Figure A2:** (a) The figure compares the root zone storage capacity ($S_r$) with the ecosystem state (i.e., tree cover). This figure expands on the conceptual illustration from Fig. A1, showing how the ecosystem's precipitation and evaporation fluxes contribute to $S_r$ under different forest transition scenarios: (b) forest-savanna transition, (c) transition to a more water-limited state, and (d) reversion to a less water-limited state.