# Peer review of "Multi-fold increase in rainforests tipping risk beyond 1.5-2°C warming"

_EGUsphere, 2023_

## Author Comment (AC1)

**Response to Reviewer #1**

1. Singh et al. classified the tropical terrestrial ecosystems under current climate and future climate by calculating the hydroclimate-derived root zone storage capacity. Then they assessed the potential rainforests tipping risk with the global warming. They found that the forest-savanna transition risk would largely increase if the climate warming is beyond 1.5-2 degrees. The topic is meaningful and interesting since the land cover change used in current ESMs of CMIP6 is lacked of the consideration of the effects of hydroclimate.

   ***Response***: We are pleased to hear that the reviewers found the research topic to be of considerable interest.

2. However, readers could be hard to follow and even confused in the main text, because some introduction of method and discussions are not easy to understand.

   ***Response:*** In the revised manuscript, we will enhance the clarity and articulation of our methods and discussion sections. We plan to refine the *'Root zone storage capacity-based framework for projecting forest transitions'* and '*Projecting forest transitions under future climate change*' subsection, a fundamental part of our methodological framework, by dividing it into several subsections for better understanding. Additionally, in the discussion section, we will clarify '*Comparing prescribed future land-use with projected transitions*' and '*Limitations and sensitivity analyses*' to address any complexities around projected and prescribed land-uses and current limitations of the ESMs in projecting tipping risk that may have made them difficult to follow.

3. More importantly, the main findings are not clearly shown in the main text. For example, in the Abstract, the "1.5-6 times" growth is the key finding for this study (also corresponding to the title), but how these values are derived is not shown.

   ***Response:*** To better convey the escalating risk of rainforest tipping under different shared socioeconomic pathways (SSPs), we articulate that "For Amazon, this risk can grow by about 1.5-6 times compared to its immediate lower warming scenario,.....". This explanation corresponds with Fig. 2, which shows the escalating risk of tipping in the Amazon rainforest (measured by area). Relative to SSP1, the risk of tipping increases sixfold under SSP2, twentyfold under SSP3, and thirty-threefold under SSP5. However, compared to their immediate lower warming scenario, the risk multiplies six times when transitioning from SSP1 to SSP2, 3.5 times from SSP2 to SSP3, and 1.5 times from SSP3 to SSP5.

   These detailed comparisons will be incorporated into the results section of the revised manuscript for enhanced clarity.

4. In this study, >20% of model convergence are regarded as 'moderate model agreement' or 'moderate-high model agreement'. Given that the findings with >20% of model convergence are important in this research, I doubt whether the 20% is too low to hardly help obtain the robust results.

   ***Response:*** In our study, a model convergence > 20%—or agreement among more than 20% of Earth System Models (ESMs)—indicates that if the same pixel across multiple models is classified similarly, it is then designated as undergoing a certain transition. For example, for a pixel to be considered part of the forest-savanna transition in the context of more than 20% model convergence, it requires the consensus of more than 7 out of 33 models confirming this transition.

   While a threshold of >20% may seem low based on the total number of ESMs analyzed, it's important to recognize the varying and often limited capabilities of these models, particularly in simulating biophysical interaction and emerging properties due to our limited understanding of the Earth system (Arora et al. 2023; Reyer et al. 2015). **Opting for a majority-based**

**consensus in ESMs could overlook critical tipping risks identified by a subset of models, which may be as likely to reveal the actual state of such risks as their counterparts.**

Therefore, recognizing these challenges in accurately modeling land surface interactions and transitions within ESM, our study showcases model agreement levels of both >20% and >50% (Fig. 2). Contrary to previous studies that either relied on a single model or used an ensemble of hydroclimate estimates from (6 to 7) ESMs for projecting tipping risks (Supplementary Table 3), which could introduce a selective bias, our approach seeks to address this concern. By integrating a broad spectrum of transitions projected by different ESMs, our model enhances transparency and offers a more comprehensive understanding of rainforest tipping risk. Through this, we aim to illuminate both the discrepancies and alignments among the projected transitions, offering a foundation for future research into the causes behind these potential variances.

We will emphasize these points in the revised manuscript.

*Arora, Vivek K., et al. "Towards an ensemble-based evaluation of land surface models in light of uncertain forcings and observations." Biogeosciences 20.7 (2023): 1313-1355.*

*Reyer, Christopher PO, et al. "Forest resilience and tipping points at different spatio-temporal scales: approaches and challenges." Journal of Ecology 103.1 (2015): 5-15.*

5. It is interesting to compare the prescribed future land-use in IAMs with the projected transitions in this study. But it is not clear for readers which results are more robust. Readers cannot figure it out from the discussions of the authors. For example, on the one hand, the author said the extent of forest-savanna transitions is often underestimated in prescribed land-use compared to those projected in their study. In this case, it seems that results from this study are regarded as more robust. However, on the other hand, the authors said forests that revert to a 'less water-stressed state' is overestimated in their analysis. It seems that results from the prescribed future land-use in IAMs are more robust.

   *Response:* The following statement will be added to the revised manuscript: "Our analysis reveals that the extent of forest-savanna transitions is often underestimated in prescribed land-use compared for South America to those projected in this study (i.e., prescribed land use predicts forests in the region that risk tipping; Fig. 5a). Furthermore, forests that revert to a 'less water-stressed state' are again underestimated for both South America and Africa (i.e., prescribed land-use predicts non-forested areas in the region that can sustain forests; Fig. 5c)."

   We will further add caution on how to interpret the results from this comparison, highlighting how prescribed land use might have introduced unrealistic hydroclimatic trends and how that could influence our comparison.

*Specific comments:*
6. Line 28: which scenario for this growth by about 1.5-6 times.
   *Response:* This comment is addressed in our response to Review #1 comment 3.

7. Lines 98-100: please explain why the hydroclimate and ecosystem can be regarded as in equilibrium. The hydroclimate and ecosystem are projected by ESM in SSP scenario simulations, which are apparently not in equilibrium because of the continued warming.
   *Response*: The aim of this research stems from a key limitation in Earth System Models (ESMs): while hydroclimate is dynamically projected, land use is statically prescribed. To assess whether current forest ecosystems will maintain their status or transition to savannas (i.e., shift between equilibrium states) by the century's end (2086-2100), we rely on the assumption that the projected hydroclimate will dictate the suitable/sustainable ecosystem

type, suggesting a balance between hydroclimate conditions and ecosystem states. This approach is necessitated by the complex and often lengthy processes required for ecosystems to reach equilibrium following disturbances (for instance, the Amazon may take 50-200 years to tip; Armstrong McKay et al. 2022). **Factors such as the severity of perturbations, mechanisms of forest decline/mortality, and adaptation play crucial roles, making the dynamic simulation of such transitions in ESMs challenging.**

**By assuming equilibrium between ecosystems and their hydroclimates, we admittedly overlook the precise temporal dynamics of rainforest tipping.** However, this trade-off allows us to identify areas at risk of tipping, providing valuable insights for developing mitigation strategies to prevent such transitions. This methodological choice enhances our understanding of potential tipping risks and could help inform proactive conservation and climate action plans.

*Armstrong McKay, David I., et al. "Exceeding 1.5 C global warming could trigger multiple climate tipping points." Science 377.6611 (2022): eabn7950.*

8. Lines 130-131: The spatial resolutions of most of ESMs output are close to 0.25 degree? I suppose that the spatial resolutions of most of ESMs are much lower than 0.25 degree.
   *Response:* The resolution of Earth System Models (ESMs) typically ranges from 1° to 1.5°, with EC-Earth3 offering the highest resolution at 0.7° and CanESM5 having the lowest at 2.8°. In the manuscript, we do state that 'Though obtained estimates from different ESMs are at different spatial resolutions, we bilinearly interpolated them to 0.25º for this analysis' [Page 4, Line 130-131].
   In the revised manuscript, we will provide detailed resolutions for all ESMs within the Supplementary Information.

9. Line 162: "to reduce loss of root zone moisture storage"?
   *Response:* Thank you for pointing this out. This will be corrected in the revised manuscript.

10. Line 183: "the actual state of the ecosystems" includes many aspects of ecosystems. "this model can capture the dynamics of actual soil moisture availability for the ecosystems" would be better.
    *Response:* Indeed, thank you for pointing this out. This will be corrected in the revised manuscript.

11. Line 380-381: please add the references of related figure(s).
    *Response:* Thank you for noticing this. The figure reference will be added in the revised manuscript.

12. Lines 590-592: But as shown in Figure 3, even in SSP1-2.6, there are still many regions belonging to "Transition to a more water-stressed state".
    *Response:* Indeed, but depending on how the model was parameterized, SSP1-2.6 still leads to approximately 1.3-2.4ºC warming. This warming is expected to not only decrease precipitation and increase precipitation seasonality but also elevate evaporation rates beyond current climate conditions. The combination of higher evaporation and reduced precipitation favors forest ecosystems that enhance their root zone storage capacity in order to ensure sufficient moisture is retained for dry spells.

---

## Author Comment (AC2)

**Response to Reviewer #2**

1. Singh et al. compare estimates of the plant-accessible root-zone water storage capacity (Sr) to the expected amount of water needed to supply ET during a 20-year return drought length across the Amazon and Congo rainforests. They classify forests with Sr smaller than the amount of storage needed to withstand a drought of this magnitude as water-stressed and compare the current extent of water-stressed forests to the projected extent based on simulated future ET and P used to generate future Sr estimates. By using thresholds of water limitation associated with the transition of ecosystems from forest to savanna from a previous publication, they identify areas that might experience forest-savanna transition.
This work is important because of our limited understanding of climate change-induced ecosystem transition.
The figures are well made and clear, with excellent explanations both in the figures themselves and in the captions. I also appreciate the attention to subsurface moisture availability as a driving factor of landcover transition and water stress.
   ***Response***: We appreciate the reviewer's positive feedback on the significance of our work and the clarity of our visualizations.

2. However, I had difficulty following the methods in this paper. I am also concerned with the interpretation of the root-zone water storage capacity metric.
I am confused by the authors' method of calculating Sr as well as their conversion of Sr to an indication of water stress. Figuring out what they were doing took me quite some time and involved reading their previous paper on this topic [1]. I am still unsure if I understand their methods and believe other readers would also have difficulty following. I would recommend improving the clarity of the terminology used in the method (for example, differentiating between 'maximum deficit' and 'Sr') as well as incorporating more of the "Calculating root zone storage capacity" section of the SI into the main text.
   ***Response:*** In the revised manuscript, we will enhance the clarity and articulation of our methods section.

   - *First, we will improve the root zone storage capacity definition and the description of the estimation method. In the revised manuscript, we will expand and move the description of $S_r$ estimation method from Supplementary Information to the main text as suggested by the reviewer. In addition, we will add a figure to visualize the $S_r$ estimation method (see further down):*

   **Root zone storage capacity ($S_r$) refers to the maximum amount of soil moisture that can be accessed by vegetation for transpiration during dry periods.** Dry periods refer to periods in which evaporation is greater than rainfall, irrespective of the seasons. The method of estimating $S_r$ is based on the mass-balance approach, through which we assume that ecosystems do not invest in expanding their rooting systems beyond the need for maximum accumulated water deficit experienced by the vegetation.

   For estimating $S_r$, we first obtained the water deficit ($D_t$) at daily time step from the daily estimates of precipitation ($P_t$) and evaporation ($E_t$) using:

   $$D_t = E_t - P_t \tag{1}$$

   Here, *t* denotes the day count since the start of the simulation, with simulation for each grid starting in the month with maximum precipitation. Second, we calculated the accumulated water deficit integrated at each one-day timestep for one year using:

$$D_{a(t+1)} = \max\{0, D_{a(t)} + D_{t+1}\}$$

(2)

Where $D_{a(t+1)}$ is the accumulated water deficit at each time step. Here, an increase in the accumulated deficit will occur when $E_t > P_t$, and a decrease when $E_t < P_t$. However, since this algorithm estimates a running estimate of root zone storage reservoir size, we use a maximum function to calculate the accumulated deficit, which by definition can never be below zero. Not allowing $D_{a(t+1)}$ to be negative also means that excess moisture from precipitation will either contribute to deep drainage or runoff. Lastly, the maximum accumulated annual water deficit ($D_{a,y}$) will represent the maximum storage required by the vegetation to respond to the critical dry periods.

$$D_{a,y} = \max\{D_{a(t+1)}\} \quad t = 1 : n-1$$

(3)

This simulation runs for a whole year, with $n$ denoting the number of days in year $y$.

Previous studies also acknowledge that different terrestrial ecosystems are adapted to different drought return periods (i.e., recurrence interval) (Wang-Erlandsson et al., 2016). However, to avoid any artificially introduced transitions between different ecosystems (i.e., forest, savanna and grasslands), a uniform 20-year drought return period is used to calculate $S_r$ (Bouaziz et al., 2020). This drought return period is then simulated for the whole time series of $D_{a,y}$ based on the Gumbel extreme value distribution (Gumbel, 1958). Thus, $S_r$ refers to the maximum amount of root zone moisture available to vegetation for transpiration during the largest accumulated water deficit expected every twenty years under static climate conditions. (See the insets 'a' and 'b' in the figure below.)

*Bouaziz, Laurène JE, et al. "Improved understanding of the link between catchment‐scale vegetation accessible storage and satellite‐derived Soil Water Index." Water Resources Research 56.3 (2020): e2019WR026365.*
*Gumbel, EJ. Statistics of extremes. Columbia University Press, New York (1958).*
*Wang-Erlandsson, Lan, et al. "Global root zone storage capacity from satellite-based evaporation." Hydrology and Earth System Sciences 20.4 (2016): 1459-1481.*

- *Second, we will replace the term 'water-stressed' with 'water-limited' in the revised manuscript to prevent confusion.*

We believe that the source of this confusion is probably the use of the term 'water stress', which in our context refers to the magnitude and duration of water-deficit such that it inhibits plant growth, as well as the probability of them transitioning to a savanna ecosystem (Singh et al. 2020). However, we acknowledge that the term 'water stress' is commonly used in various contexts with different meanings, which could potentially confuse readers.

**In this revision, we would therefore like to replace 'water-stress' with 'water-limited' to more precisely describe the effects of shifting hydroclimatic conditions on forest ecosystems, specifically in terms of inhibiting plant growth and the potential of them approaching the threshold of forest-savanna transition.** Essentially, as ecosystems accumulate water deficits—resulting from higher evaporation, reduced precipitation, or extended dry spells—they adapt by enhancing their $S_r$, extending their roots to tap into deeper soil moisture for transpiration in dry periods. However, there is a limit to how much $S_r$ can compensate for, beyond which further hydroclimatic shifts may lead to ecosystem tipping to savanna. See the inset 'c-f' in the figure below.

We will adapt the manuscript to this 'water-limited' term accordingly.

[Figure]

**Fig.** Conceptual framework to assess forest transitions using root zone storage capacity ($S_r$). (a) We illustrate the ecosystem's root zone and associated water storage, and (b) explain how we employ a mass-balance method, utilizing precipitation and evaporation data, to compute the accumulated water deficit and its relation to $S_r$. (c) Here, we outline how variations in $S_r$ lead to forest transitions, with subsequent panels (d-f) offering detailed

insights into the associated changes in the ecosystem's above-ground structure and below-ground root zone storage capacity. This figure in panel (b) is adapted from Singh et al. 2023.

3. If I am understanding the authors' calculation of Sr correctly, then I am skeptical about their interpretation of it. This confusion starts for me in the first sentence of the abstract. Forests themselves don't "store moisture" - the subsurface may store moisture (abiotically) and rainforests can access this moisture via roots. There are many places in the manuscript (for example, line 47) where the authors do not fully articulate the abiotic influence on Sr, and I think this may have large consequences for their interpretation of Sr as an indication of water stress.

*Response:* This first sentence will be rephrased as **'Tropical rainforests invest in their root systems to access soil moisture stored in their root zone from water-rich periods for use during water-scarce periods'**. We will address and correct these discrepancies across the manuscript.

Sentences in Line 47 will be rephrased as 'Since the rooting structure is challenging to measure at the ecosystem scale, previous studies have found that empirical mass balance-derived root zone storage capacity ($S_r$) correlates well with ecosystems' capacity to access water in its root zone and modify their above-ground structure accordingly; and thus can serve as a proxy for assessing an ecosystem's capacity to utilize subsurface moisture in maintaining its structure and functions under future climate change scenarios'.

We wish to highlight that, in theory, **the empirical mass-balance-based estimates of $S_r$ should reflect the impact of abiotic factors on vegetation's ability to access subsoil moisture.** This is because these mass-balance-based estimates are based on actual evaporation and precipitation data, reflecting the hydrological capacity of the root zone, albeit without directly accounting for the specific rooting strategies of the ecosystem. Nevertheless, these estimates encompass adjustments to rooting strategies influenced by, for example, soil type and structure, nutrient availability, oxygen levels, and mechanical resistance (such as rocks or dense layers that may impede growth and access to soil moisture). For instance, if a forest ecosystem cannot access sufficient soil moisture from deeper layers (i.e., water is not the limiting factor), it may extend its roots laterally to seek moisture; or will transition to a state that doesn't have high evaporative demand (i.e., ecosystem's investment in expanding their root zone is restricted by local factors). In the revised manuscript, we will expand on these factors in more detail.

4. For example, in my understanding, having a large deficit does not necessarily translate to water stress. Instead, it is a reflection of ET from storage, and therefore, of an ecosystem having a high Sr. Whether this high Sr confers resilience or risk to an ecosystem cannot be known from the Sr estimate itself. For example, "excessive short-term water deficits" (line 53-60) can be rephrased as 'a lot of ET and not a lot of P' which may mean that vegetation has a lot of access to subsurface water, not that it is at risk of mortality, as the authors write. I think this paper may be conflating drought and low subsurface moisture levels with the deficit when in my understanding the deficit and Sr are very difficult to convert directly to mortality or stress metrics (see [2] as an example of how complex using deficit-based methods to understand landcover can be). If my understanding of the methods of this paper is correct, this conflation would be a fundamental misuse of Sr.

*Response:* This response is partially covered in our previous response to comment 2.
Indeed, large water-deficits do not automatically imply water stress (as per its traditional definition), but rather indicate a high $S_r$. **However, our previous studies have shown that, under further episodic changes to their hydroclimate (e.g., decrease in precipitation, increase in evaporation or longer dry periods due to a warmer climate), tropical forest**

ecosystems with a higher $S_r$ tend to be more susceptible to transitioning into savanna ecosystems than those with a lower $S_r$ (Singh et al. 2020; 2022). This is because every ecosystem has evolved to sustain its structure and functions under specific hydroclimatic conditions. When an ecosystem exhibits a root zone storage capacity at the upper limits for its type, additional drying of the hydroclimate could trigger a transition from forest to savanna ecosystems. This shift occurs because savanna ecosystems are inherently more adapted and competitive under drier hydroclimatic conditions. In this study, our goal is to apply and extend these empirical insights to evaluate the resilience of rainforest ecosystems and their tipping risk under future climate change conditions.

Therefore, we would like to replace 'water-stress' with 'water-limited' to more precisely describe the effects of shifting hydroclimatic conditions on forest ecosystems, specifically in terms of inhibiting plant growth and the potential of them approaching the threshold of forest-savanna transition. For more information on how $S_r$ links to forest transition and tipping risk, please check our conceptual figure above, which we will incorporate in our revised manuscript.

We also wish to highlight that Hahm et al. (2019; *study recommended by the reviewer*), discuss how the differences in plant-available water (or root zone available water) primarily depend on the depth and degree of weathering in the bedrock beneath the soils (i.e., part of critical zones), which impacts its water retention capacity. **Our $S_r$ metric is designed to quantify the hydrological storage capacity within the root zone exclusively, without delving into rooting extent (e.g., depth or structure), which is influenced by multiple factors beyond the scope of this metric.** However, we would like to mention that apart from our studies, **the hydrological capacity of the root zone, derived from a mass-balance (or climate-based) approach, has been extensively examined, providing insights that more accurately reflect both hydrological regimes (de Boer-Euser et al. 2016) and vegetation dynamics (Dralle et al. 2020; Gao et al. 2014).** Notably, Dralle et al. (2020) assert that this mass-balance methodology can infer attributes of the critical zone, including characteristics of deeper weathered bedrock beneath superficial soil layers, a concept further investigated by McCormick et al. (2021).

*Dralle, David N., et al. "Plants as sensors: vegetation response to rainfall predicts root-zone water storage capacity in Mediterranean-type climates." Environmental Research Letters 15.10 (2020): 104074.*

*McCormick, Erica L., et al. "Widespread woody plant use of water stored in bedrock." Nature 597.7875 (2021): 225-229.*

*Gao, Hongkai, et al. "Climate controls how ecosystems size the root zone storage capacity at catchment scale." Geophysical Research Letters 41.22 (2014): 7916-7923.*

*Hahm, W. Jesse, et al. "Lithologically controlled subsurface critical zone thickness and water storage capacity determine regional plant community composition." Water Resources Research 55.4 (2019): 3028-3055.*

*Singh, Chandrakant, et al. "Rootzone storage capacity reveals drought coping strategies along rainforest-savanna transitions." Environmental Research Letters 15.12 (2020): 124021.*

*Singh, Chandrakant, et al. "Hydroclimatic adaptation critical to the resilience of tropical forests." Global Change Biology 28.9 (2022): 2930-2939.*

*de Boer-Euser, Tanja, et al. "Influence of soil and climate on root zone storage capacity." Water Resources Research 52.3 (2016): 2009-2024.*

5.  At present it is difficult to tell how much Sr is being misinterpreted because it is difficult to follow the methods of the paper. These issues could possibly be improved by (1) improving the clarity of the methods so it is possible for a reader to follow exactly how the metrics are calculated and compared to one another to derive the categories plotted in the figures, (2) more discussion on the abiotic controls of the deficit and how that might conflate interpretations of low vs high Sr as indicative of water stress and landcover transition, including careful examination and explanation of the logic outlined in Figure 2a (describing the relationship between Sr and transition) and (3) more information on the authors' previous thresholds for forest-savanna transition, which are critical to accepting their main results here.
    ***Response:*** We agree with the reviewer's suggestions and, as outlined in our previous responses, intend to enhance the clarity of our methods section and its connection to our results. To this end, we will introduce a dedicated sub-section detailing the findings from our prior study and describing how we extend these insights to forecast forest transitions under future climate scenarios.

*Other comments:*
6.  Line 137: Using a threshold of 50% to determine if a pixel should be classified as "forest" seems generous, especially as other ecosystems might be expected to have very different ET (e.g. crops) and would therefore alter estimates of Sr to a large extent. It might be helpful to see the distribution of fractional forest cover present in your "forest" pixels or to otherwise show that the most of the area you are analyzing is more fully forested than 50% coverage.
    ***Response:*** It is widely accepted to use a threshold exceeding 50% to differentiate natural forests, characterized by woody vegetation cover greater than 50%, from savannas, where woody vegetation cover is 50% or less (Staal et al. 2018; Zemp et al. 2017). However, we acknowledge that in some instances, evaporation from non-forest land use could affect the overall evaporative trends of a pixel, especially as we aggregate the dataset to a coarser resolution.

    Following the reviewer's suggestion, we present data on the fraction of forest cover within a pixel (*figure below*). Our analysis indicates that instances of forest pixels containing land uses typically associated with non-forest areas frequently arise at the interface of natural and human-influenced regions. However, a visual comparison of these forested areas with the transition zones identified in our study (Fig. 3) reveals that they do not appear to significantly influence the overall findings.

[Figure]

[Figure]

Forest defined by Globcover at 300m resolution

Forest after interpolation to 0.25° resolution
*Fraction of forest within a pixel*

[Figure]

50%                    100%

*Staal, Arie, et al. "Forest-rainfall cascades buffer against drought across the Amazon." Nature Climate Change 8.6 (2018): 539-543.*
*Zemp, Delphine Clara, et al. "Self-amplified Amazon forest loss due to vegetation-atmosphere feedbacks." Nature communications 8.1 (2017): 14681.*

7. The model agreement threshold of 20% seems too low for gaining a robust understanding of likely future transitions. This is especially concerning as the area with >50% model agreement is so small (for example in Figure 2, the forest-savanna transition in Africa).

**Response:** While a threshold of >20% may seem low based on the total number of ESMs analyzed, it's important to recognize the varying and often limited capabilities of these models, particularly in simulating biophysical interaction and emerging properties due to our limited understanding of the Earth system (Arora et al. 2023; Reyer et al. 2015). **Opting for a majority-based consensus in ESMs could overlook critical tipping risks identified by a subset of models, which may be as likely to reveal the actual state of such risks as their counterparts.**

Therefore, recognizing these challenges in accurately modeling land surface interactions and transitions within ESM, our study showcases model agreement levels of both >20% and >50% (Fig. 2). Contrary to previous studies that either relied on a single model or used an ensemble of hydroclimate estimates from (6 to 7) ESMs for projecting tipping risks (Supplementary Table 3), which could introduce a selective bias, our approach seeks to address this concern. By integrating a broad spectrum of transitions projected by different ESMs, our model enhances transparency and offers a more comprehensive understanding of rainforest tipping risk. Through this, we aim to illuminate both the discrepancies and alignments among the projected transitions, offering a foundation for future research into the causes behind these potential variances.

*Arora, Vivek K., et al. "Towards an ensemble-based evaluation of land surface models in light of uncertain forcings and observations." Biogeosciences 20.7 (2023): 1313-1355.*

*Reyer, Christopher PO, et al. "Forest resilience and tipping points at different spatio-temporal scales: approaches and challenges." Journal of Ecology 103.1 (2015): 5-15.*

8. Line 48: The '-' is a typo? This sentence doesn't make sense.
   **Response:** Yes, this is a typo. We appreciate your attention in highlighting this. The corrected sentence can be found in our response to the reviewer's comment 3.

9. Typo in Figure 2 panel a, under 'transition to a more water-stressed state" (hydraulic failures).
   **Response:** Thank you for bringing this to our attention. This will be corrected in the revised manuscript.

---

## Author Response (AR1)

**Multi-fold increase in rainforests tipping risk beyond 1.5-2ºC warming**

**Response to Reviewer #1**

1. Singh et al. classified the tropical terrestrial ecosystems under current climate and future climate by calculating the hydroclimate-derived root zone storage capacity. Then they assessed the potential rainforests tipping risk with the global warming. They found that the forest-savanna transition risk would largely increase if the climate warming is beyond 1.5-2 degrees. The topic is meaningful and interesting since the land cover change used in current ESMs of CMIP6 is lacked of the consideration of the effects of hydroclimate.

   ***Response***: We are pleased to hear that the reviewers found the research topic to be of considerable interest.

2. However, readers could be hard to follow and even confused in the main text, because some introduction of method and discussions are not easy to understand.

   ***Response:*** Thank you for your valuable feedback regarding the clarity of our manuscript.

   In the revised manuscript, we enhance the clarity and articulation of our Methods and Discussion sections [Pg 5-10, Ln 153-304; Pg 14-20, Ln 403-535].

   To enhance the readability of our Methodology section, we have restructured the 'Root zone storage capacity-based framework for projecting forest transitions' into three distinct subsections:
   i. "**Estimating mass-balance derived root zone storage capacity ($S_r$)**", which outlines the methods for calculating root zone storage capacity, [Pg 6, Ln 165-191]
   ii. "**Determining root zone storage capacity thresholds for forest transitions**", providing the rationale for the ecosystem classifications and $S_r$ thresholds suggested by Singh et al. (2020), and [Pg 7-8, Ln 193-250]
   iii. "**Projecting forest transitions under future climate change**", detailing the use of empirical and CMIP6 hydroclimate data to project forest transitions. [Pg 8-9, Ln 252-290]

   Additionally, we have enhanced the clarity and understandability of all subsections within the Discussion. [Pg 14-20, Ln 403-535]

   *Singh, Chandrakant, et al. "Rootzone storage capacity reveals drought coping strategies along rainforest-savanna transitions." Environmental Research Letters 15.12 (2020): 124021.*

3. More importantly, the main findings are not clearly shown in the main text. For example, in the Abstract, the "1.5-6 times" growth is the key finding for this study (also corresponding to the title), but how these values are derived is not shown.

   ***Response:*** We apologise that the key findings of the study were not apparent from our abstract and results.

   In the revised manuscript, we clarify that:
   - **In Abstract**: "Furthermore, warming beyond 1.5-2ºC will significantly elevate the risk of a forest-savanna transition. In the Amazon, the forest area at risk of such a transition grows by about 1.7-5.8 times in size compared to their immediate lower warming scenario (e.g., SSP2-4.5 compared to SSP1-2.6). In contrast, the risk growth in the Congo is less substantial, ranging from 0.7-1.7 times." [Pg 1, Ln 31-34]

- **In the Results section**: "When comparing the changes in forest-savanna transition risk areas relative to their immediate lower warming scenarios, we find considerable increases for South America. The highest relative growth of approximately 5.75 times is observed between SSP1 and SSP2, with the forest area under risk increasing from $0.04 \times 10^6$ km$^2$ to $0.23 \times 10^6$ km$^2$, respectively. It increases by 3.48 times from SSP2 to SSP3 ($0.23 \times 10^6$ km$^2$ to $0.80 \times 10^6$ km$^2$), and by 1.65 times from SSP3 to SSP5 ($0.80 \times 10^6$ km$^2$ to $1.32 \times 10^6$ km$^2$). For Africa, however, the increases are more modest: the risk grows by 1.29 times from SSP1 to SSP2 ($0.17 \times 10^6$ km$^2$ to $0.22 \times 10^6$ km$^2$), by 1.63 times from SSP2 to SSP3 ($0.22 \times 10^6$ km$^2$ to $0.36 \times 10^6$ km$^2$), and is observed to decrease by 0.72 times from SSP3 to SSP5 ($0.36 \times 10^6$ km$^2$ to $0.26 \times 10^6$ km$^2$)." [Pg 10-11, Ln 324-331]

4. In this study, >20% of model convergence are regarded as 'moderate model agreement' or 'moderate-high model agreement'. Given that the findings with >20% of model convergence are important in this research, I doubt whether the 20% is too low to hardly help obtain the robust results.

*Response:* While a threshold of >20% may seem low given the total number of ESMs analysed, it is important to recognise the variable and often limited capabilities of these ESMs, particularly in simulating biophysical interaction and emerging properties due to our limited understanding of the Earth system (Lenton et al., 2019; Stevens and Bony, 2013). **Opting for a majority-based consensus in ESMs could overlook critical tipping risks identified by a minority of models, which might provide insights as valid as those from more widely agreeing models** (Arora et al., 2023; Reyer et al., 2015)**. [Pg 9, Ln 282-290]**

Recognising these challenges in accurately modelling land surface interactions and transitions within ESM, our study showcases model agreement levels of both >20% and >50% (Fig. 2). Contrary to previous studies that either relied on a single model or used an ensemble of hydroclimate estimates from 6 to 7 ESMs for projecting tipping risks (Table S3), which could introduce a selective bias, our approach seeks to address this concern.

We believe that by considering simulations from multiple ESMs under different SSP scenarios, not only do we highlight the agreements and conflicts between potential transitions; but also allow future studies to disentangle vegetation-climate feedbacks and improve the modelling of local-scale interactions (e.g., vegetation's water-uptake profile, species response to $CO_2$ fertilisation) in the ESMs. [Pg 20, Ln 531-535]

In the revised manuscript, the elements from above-mentioned paragraphs are added to the Methodology subsection '**Projecting forest transitions under future climate change**', and the Discussion subsection '**Limitations**'.

Thank you for bringing this to our attention.

*Arora, Vivek K., et al. "Towards an ensemble-based evaluation of land surface models in light of uncertain forcings and observations." Biogeosciences 20.7 (2023): 1313-1355.*

*Lenton, Timothy M., et al. "Climate tipping points—too risky to bet against." Nature 575.7784 (2019): 592-595.*

*Reyer, Christopher PO, et al. "Forest resilience and tipping points at different spatio-temporal scales: approaches and challenges." Journal of Ecology 103.1 (2015): 5-15.*

*Stevens, Bjorn, and Sandrine Bony. "What are climate models missing?." science 340.6136 (2013): 1053-1054.*

5. It is interesting to compare the prescribed future land-use in IAMs with the projected transitions in this study. But it is not clear for readers which results are more robust. Readers cannot figure it out from the discussions of the authors. For example, on the one hand, the author said the extent of forest-savanna transitions is often underestimated in prescribed land-use compared to those projected in their study. In this case, it seems that results from this study are regarded as more robust. However, on the other hand, the authors said forests that revert to a 'less water-stressed state' is overestimated in their analysis. It seems that results from the prescribed future land-use in IAMs are more robust.

   *Response:* Thank you for your feedback on the clarity of this comparison.

   In the revised manuscript, these sentences in question have been revised as follows: "The most noticeable discrepancies are observed in South America, where the extent of forest-savanna transitions is underestimated in prescribed land-use scenarios compared to those projected in this study **(i.e., prescribed land-use predicts forests in the region whose hydroclimate can't support forest; Fig. 4 and 5a).** Additionally, in South America, our analysis highlights the potential of some forests reverting to a 'less water-limited state' in places where the prescribed land-use in the ESMs suggest non-forest landscape (Fig. 4 and 5c). These discrepancies arise because the prescribed land-use in CMIP6-ESMs do not shift in response to hydroclimatic changes. Despite our approach assuming equilibrium and overlooking the temporal dynamics of transitions, based on broad climate change pattern (Sect 4.2), we believe it more accurately represents the ecohydrological state of the ecosystems". [Pg 17-18, Ln 479-487]

   Furthermore, we have also added caution since prescribed land-use can influence biophysical processes in ESMs.
   "**However, these prescribed land-uses can introduce errors in subsequent biophysical processes simulated in ESMs (Ma et al., 2020), affecting the accuracy of projected transitions.** For example, prescribing a region as a forest that would be grassland in the future will lead to the extraction of deeper subsoil moisture in ESMs, which (actual) grasslands do not have the capacity to access (Ahlström et al., 2017; Yu et al., 2022). This will result in an overestimation of the ecosystem's evaporation, potentially altering precipitation patterns downwind and leading to inaccurate water budget assessments for these ecosystems. Consequently, causing erroneous projections of the ecosystem state. These discrepancies underscore the urgent need for enhancements in the land surface components of ESMs, enabling dynamic simulations of vegetation-climate feedbacks. Such improvements would provide a more accurate representation of the ecohydrology of terrestrial ecosystems and their response to changing climate conditions." [Pg 18, Ln 488-497]

*Specific comments:*
6. Line 28: which scenario for this growth by about 1.5-6 times.
   *Response:* This comment is addressed in our response to comment 3.

7. Lines 98-100: please explain why the hydroclimate and ecosystem can be regarded as in equilibrium. The hydroclimate and ecosystem are projected by ESM in SSP scenario simulations, which are apparently not in equilibrium because of the continued warming.
   *Response*: We acknowledge that the ecosystem and hydroclimate might not be in an actual equilibrium state in ESMs. Consequently, we explicitly state the following:
   "However, we do not account for the time required for ecosystems to reach their (long-term) equilibrium state, which previous studies suggest can take between 50-200 years after crossing the tipping point (Armstrong McKay et al., 2022). " [Pg 3, Ln 96-98]

Thank you for bringing this to our attention.

*Armstrong McKay, David I., et al. "Exceeding 1.5 C global warming could trigger multiple climate tipping points." Science 377.6611 (2022): eabn7950.*

8. Lines 130-131: The spatial resolutions of most of ESMs output are close to 0.25 degree? I suppose that the spatial resolutions of most of ESMs are much lower than 0.25 degree.
   *Response:* The resolution of Earth System Models (ESMs) typically ranges from 1° to 1.5°, with EC-Earth3 offering the highest resolution at 0.7° and CanESM5 having the lowest at 2.8°.

   In the Methods section of the manuscript, we do state that "Though obtained estimates from different ESMs are at different spatial resolutions, we bilinearly interpolated them to 0.25º for this analysis". [Pg 4, Ln 127-128]

   In the revised Supplementary Information, Table S1 now includes details about the spatial resolutions for all analysed ESMs. [Pg 26-28]

9. Line 162: "to reduce loss of root zone moisture storage"?
   *Response:* Thank you for pointing this out.

   This has been corrected in the revised manuscript:
   "Furthermore, forest ecosystems adapt to climate change by optimising water distribution through mechanisms such as hydraulic redistribution (Liu et al., 2020; Oliveira et al., 2005), enhancing water-use efficiency by regulating stomatal conductance, and even shredding leaves (Wolfe et al., 2016) **to minimise moisture loss** (Barros et al., 2019; Brum et al., 2019; Lammertsma et al., 2011)." [Pg 2, Ln 54]

10. Line 183: "the actual state of the ecosystems" includes many aspects of ecosystems. "this model can capture the dynamics of actual soil moisture availability for the ecosystems" would be better.
    *Response:* Indeed, thank you for pointing this out. This has been corrected in the revised manuscript. [Pg 6, Ln 178]

11. Line 380-381: please add the references of related figure(s).
    *Response:* Thank you for noticing this. The figure reference has been added to the revised manuscript. [Pg 12, Ln 369]

12. Lines 590-592: But as shown in Figure 3, even in SSP1-2.6, there are still many regions belonging to "Transition to a more water-stressed state".
    *Response:* Indeed, but depending on how the model was parameterised, SSP1-2.6 still leads to approximately 1.3-2.4ºC warming. This warming is expected to not only decrease precipitation and increase precipitation seasonality, but also elevate evaporation rates (ecosystem water demand) beyond current climate conditions. The combination of higher evaporation and reduced precipitation favours forest ecosystems that enhance their root zone storage capacity in order to ensure sufficient moisture is accessible for transpiration during dry spells.

**Multi-fold increase in rainforests tipping risk beyond 1.5-2ºC warming**

**Response to Reviewer #2**

1. Singh et al. compare estimates of the plant-accessible root-zone water storage capacity (Sr) to the expected amount of water needed to supply ET during a 20-year return drought length across the Amazon and Congo rainforests. They classify forests with Sr smaller than the amount of storage needed to withstand a drought of this magnitude as water-stressed and compare the current extent of water-stressed forests to the projected extent based on simulated future ET and P used to generate future Sr estimates. By using thresholds of water limitation associated with the transition of ecosystems from forest to savanna from a previous publication, they identify areas that might experience forest-savanna transition.
This work is important because of our limited understanding of climate change-induced ecosystem transition.
The figures are well made and clear, with excellent explanations both in the figures themselves and in the captions. I also appreciate the attention to subsurface moisture availability as a driving factor of landcover transition and water stress.
***Response***: We appreciate the reviewer's positive feedback on the significance of our work and the clarity of our visualisations.

2. However, I had difficulty following the methods in this paper. I am also concerned with the interpretation of the root-zone water storage capacity metric.
I am confused by the authors' method of calculating Sr as well as their conversion of Sr to an indication of water stress. Figuring out what they were doing took me quite some time and involved reading their previous paper on this topic [1]. I am still unsure if I understand their methods and believe other readers would also have difficulty following. I would recommend improving the clarity of the terminology used in the method (for example, differentiating between 'maximum deficit' and 'Sr') as well as incorporating more of the "Calculating root zone storage capacity" section of the SI into the main text.
***Response:*** Thank you for your valuable feedback regarding the clarity of our methods section. In the revised manuscript, we enhance the clarity and articulation of our Methods section [Pg 5-10, Ln 153-304].

 ⇒ First, we improve the readability and interpretation of our methods. For that, we have restructured the 'Root zone storage capacity-based framework for projecting forest transitions' into three distinct subsections:
   i. "**Estimating mass-balance derived root zone storage capacity ($S_r$)**", which outlines the methods for calculating root zone storage capacity, [Pg 6, Ln 165-191]
   ii. "**Determining root zone storage capacity thresholds for forest transitions**", providing the rationale for the ecosystem classifications and $S_r$ thresholds suggested by Singh et al. (2020), and [Pg 7-8, Ln 193-250]
   iii. "**Projecting forest transitions under future climate change**", detailing the use of empirical and CMIP6 hydroclimate data to project forest transitions. [Pg 8-9, Ln 252-290]

 ⇒ Second, we improve the root zone storage capacity ($S_r$) definition and the description of its estimation. In the revised manuscript, we have added:

"Derived using the mass-balance approach, $S_r$ **represents the maximum amount of soil moisture accessed by vegetation for transpiration** (Singh et al., 2020; Wang-Erlandsson et al., 2016). **This methodology calculates the maximum extent of soil moisture within the reach of plant roots, assuming that ecosystems do not invest in expanding their root-zone storage beyond what is necessary to bridge the maximum (accumulated) water-deficit experienced by the vegetation during dry periods** (i.e., periods in which evaporation is greater than rainfall, irrespective of the seasons). **This maximum annual accumulated water deficit ($D_{a,y}$) experienced by the ecosystem is calculated using daily precipitation and evaporation estimates (Appendix A1 and Fig. A1).** Subsoil moisture beyond the reach of plant roots is primarily controlled by gravity-induced gradients (de Boer-Euser et al., 2016) and is not available for transpiration. The rationale is that any extensive investment (i.e., more than necessary) in root expansion would require carbon allocation and, thus, is inefficient from the perspective of the plants (Gao et al., 2014; Schenk, 2008). Since, this approach does not rely on prior information about vegetation, soil, or land cover-based, by using empirical (observation-based) datasets (Appendix A1 and Fig. A1), we capture the dynamics of actual soil moisture available for the ecosystems (Wang-Erlandsson et al., 2016). **The detailed methodology for calculating $S_r$ using precipitation and evaporation estimates is outlined in Appendix A1.**" [Pg 6, Ln 166-180]

Furthermore, following the reviewer's suggestion, we have also moved the $S_r$ calculation from Supplementary Information to Appendix A1 of the main text. [Pg 21-23, Ln 554-607]

We have also added a figure to visualise the conceptual understanding of $S_r$ and its estimation (Fig. A1; *see below*). [Pg 23, Ln 602-607]

In Appendix A1 and Fig. A1, we clearly delineate the differences between 'daily water deficit', 'maximum accumulated annual water deficit', 'root-zone' and $S_r$.

[Figure]

**Figure A1:** The figure illustrates the root zone storage capacity ($S_r$) of the ecosystem. (a) We show the difference between the ecosystem's root zone and how that constitutes its $S_r$. (b) Conceptual illustration of how the ecosystem's precipitation and evaporation fluxes constitute the **maximum accumulated annual water deficit ($D_{a,y}$) and $S_r$**. The figure is adapted from Singh (2023) and Wang-Erlandsson et al. (2016).

⇒ Third, in the Methods subsection **"Determining root zone storage capacity thresholds for forest transitions"** [Pg 7-8, Ln 193-250], we justify why $S_r$ can be used to indicate water stress.

We believe that the source of this confusion is probably the use of the term 'water stress', which in our context refers to the magnitude and duration of water-deficit such that it inhibits plant growth, as well as the probability of them transitioning to a savanna ecosystem (Singh et al. 2020). However, we acknowledge that the term 'water stress' is commonly used in various contexts with different meanings, which could potentially confuse readers.

To avoid further confusion in the revised manuscript, **we have now replaced 'water-stressed state' with 'water-limited state' that hopefully more precisely describes the effects of hydroclimatic conditions on forest ecosystems. The new term 'water limited', as we mean it, terms the inhibition of plant growth based on subsoil moisture availability and the potential of them approaching the threshold of forest-savanna transition.** [Pg, Ln 196-199]

The new term was chosen essentially, because as ecosystems accumulate water deficits—resulting from higher evaporation, reduced precipitation, or extended dry spells—they adapt by enhancing their $S_r$, extending their roots to tap into deeper soil moisture for transpiration in dry periods. **However, there is a limit to how much $S_r$ can compensate for, beyond which further hydroclimatic shifts may lead to forest ecosystems tipping to savanna.** Please see our explanation of 'highly water-limited forest'. [Pg 7-8, Ln 217-230]

In the revised manuscript, we have additionally included a figure to visually depict the conceptual changes in $S_r$ under future climate change, resulting from alterations in the ecosystem's evaporation and precipitation fluxes, as well as their impact on forest transitions (Fig. A2; *see below*). [Pg 25]

[Figure]

**Figure A2:** (a) The figure compares the root zone storage capacity ($S_r$) with the ecosystem state (i.e., tree cover). This figure expands on the conceptual illustration from Fig. A1, showing how the ecosystem's precipitation and evaporation fluxes contribute to $S_r$ under different forest transition scenarios: (b) forest-savanna transition, (c) transition to a more water-limited state, and (d) reversion to a less water-limited state.

3. If I am understanding the authors' calculation of Sr correctly, then I am skeptical about their interpretation of it. This confusion starts for me in the first sentence of the abstract. Forests themselves don't "store moisture" - the subsurface may store moisture (abiotically) and rainforests can access this moisture via roots. There are many places in the manuscript (for example, line 47) where the authors do not fully articulate the abiotic influence on Sr, and I

think this may have large consequences for their interpretation of Sr as an indication of water stress.

*Response:* Thank you for your feedback.

We fully agree with your reflection that forests indeed **do not store water themselves.** We apologize for any confusion caused by our previous wording and have made corrections throughout the manuscript to clarify that **'storage' refers to the roots of vegetation accessing soil moisture, rather than direct storage by the vegetation itself.**
**While we agree that the term 'root zone storage capacity' is somewhat sub-optimal, we have chosen to retain it as it is widely used and recognized within the field, and, so far, no better or more widely accepted alternative has been proposed** [Please see following references: de Boer-Euser et al., 2016; Dralle et al., 2020; Gao et al., 2014; McCormick et al., 2021; Stocker et al., 2023].

In the revised manuscript, the sentence in the first sentence in the abstract has been improved:
"Tropical rainforests rely on their root systems to access moisture stored in soil during wet periods for use during dry periods." [Pg 1, Ln 19-20]

In the revised manuscript, sentences in Line 47 (of the previous manuscript) have been improved as:
"Despite their critical role, the dynamic influence of climate change on vegetation's rooting structure and subsoil moisture is challenging to measure at the ecosystem scale (Fan et al., 2017). Thus, understanding how moisture from wet periods is stored, transmitted, and lost from soil, and how it is accessed by vegetation during dry periods, is critical to the ecohydrology and resilience of terrestrial ecosystems under climate change." [Pg 2, Ln 55-58]

We also wish to highlight that, derived using precipitation and evaporation fluxes, "$S_r$ quantifies the hydrological buffer necessary for an ecosystem to maintain its structure and functions, reflecting the amount of root zone soil moisture available to vegetation for transpiration. **Our mass-balance-based $S_r$ methodology, while not directly distinguishing between the biotic and abiotic influences on soil moisture and root characteristics, does incorporate their critical role in shaping the ecohydrology of the ecosystem under climate change.** By utilising empirical precipitation and evaporation data, our approach theoretically captures the combined impact of these biotic and abiotic factors on the actual hydrological regime (including soil moisture) of the ecosystem (Sect. 2.3.2)." [Pg 23, Ln 610-616]

In the revised manuscript, we have added the preceding paragraph to Appendix A2, titled "**Abiotic and biotic factors influence soil moisture availability**". In this subsection, we further explore the impact of biotic and abiotic factors on soil moisture availability. [Pg 23-24, Ln 610-634]

*de Boer-Euser, Tanja, et al. "Influence of soil and climate on root zone storage capacity." Water Resources Research 52.3 (2016): 2009-2024.*

*Dralle, David N., et al. "Plants as sensors: vegetation response to rainfall predicts root-zone water storage capacity in Mediterranean-type climates." Environmental Research Letters 15.10 (2020): 104074.*

*Gao, Hongkai, et al. "Climate controls how ecosystems size the root zone storage capacity at catchment scale." Geophysical Research Letters 41.22 (2014): 7916-7923.*

*McCormick, Erica L., et al. "Widespread woody plant use of water stored in bedrock." Nature 597.7875 (2021): 225-229.*

*Stocker, Benjamin D., et al. "Global patterns of water storage in the rooting zones of vegetation." Nature geoscience 16.3 (2023): 250-256.*

4. For example, in my understanding, having a large deficit does not necessarily translate to water stress. Instead, it is a reflection of ET from storage, and therefore, of an ecosystem having a high Sr. Whether this high Sr confers resilience or risk to an ecosystem cannot be known from the Sr estimate itself. For example, "excessive short-term water deficits" (line 53-60) can be rephrased as 'a lot of ET and not a lot of P' which may mean that vegetation has a lot of access to subsurface water, not that it is at risk of mortality, as the authors write. I think this paper may be conflating drought and low subsurface moisture levels with the deficit when in my understanding the deficit and Sr are very difficult to convert directly to mortality or stress metrics (see [2] as an example of how complex using deficit-based methods to understand landcover can be). If my understanding of the methods of this paper is correct, this conflation would be a fundamental misuse of Sr.

**Response:** Thank you for your feedback.

Indeed, large water deficits do not automatically imply water stress (as per its traditional definition, which has now been changed to 'water limitation'; see our response to comment 2), but rather indicate a high $S_r$. **However, our previous studies have shown that, under further episodic changes to their hydroclimate (e.g., decrease in precipitation, increase in evaporation or longer dry periods due to a warmer climate), tropical forest ecosystems with a higher $S_r$ tend to be more susceptible to transitioning into savanna ecosystems than those with a lower $S_r$ (Singh et al. 2020; 2022).**

This is because the forest ecosystem has evolved to sustain its structure and functions under specific hydroclimatic conditions. **When a forest ecosystem exhibits $S_r$ at the upper limits for its type (i.e., indicating the limit of its subsoil investment for soil moisture accessibility), further drying of the hydroclimate could trigger critical feedbacks (such as forest mortality due to hydraulic failures or ecosystem thinning due to increased fire risks). Under these conditions, a forest-savanna transition becomes likely. This shift is facilitated because savanna ecosystems are inherently better adapted to drier hydroclimatic conditions and are more fire-tolerant, making them more competitive in such environments.**

The confusion also likely arose from our use of the term 'maximum annual accumulated water deficit' and 'water-deficit' interchangeably (clarified in Appendix A1 and Fig. A1). We have corrected this throughout the revised manuscript.

We have also removed the sentence '...excessive short-term water deficits...' from the revised manuscript, as it did not effectively convey our intended message.

In the Methods subsection "**Determining root zone storage capacity thresholds for forest transitions**" of the revised manuscript, we have succinctly described how $S_r$-based forest classification is linked to forest-savanna transition risks. This includes an explanation of the potential drought coping mechanisms employed by forests under varying levels of water limitations. [Pg 7-8, Ln 193-250]

*Singh, Chandrakant, et al. "Rootzone storage capacity reveals drought coping strategies along rainforest-savanna transitions." Environmental Research Letters 15.12 (2020): 124021.*

*Singh, Chandrakant, et al. "Hydroclimatic adaptation critical to the resilience of tropical forests." Global Change Biology 28.9 (2022): 2930-2939.*

5. At present it is difficult to tell how much Sr is being misinterpreted because it is difficult to follow the methods of the paper. These issues could possibly be improved by (1) improving the clarity of the methods so it is possible for a reader to follow exactly how the metrics are calculated and compared to one another to derive the categories plotted in the figures, (2) more discussion on the abiotic controls of the deficit and how that might conflate interpretations of low vs high Sr as indicative of water stress and landcover transition, including careful examination and explanation of the logic outlined in Figure 2a (describing the relationship between Sr and transition) and (3) more information on the authors' previous thresholds for forest-savanna transition, which are critical to accepting their main results here.
**Response:** We thank the reviewer for providing clear guidance to help improve our manuscript.
In the revised manuscript, we have improved the three points mentioned by the reviewer, as outlined in more detail in our responses to comment 2, 3 and 4.

*Other comments:*

6. Line 137: Using a threshold of 50% to determine if a pixel should be classified as "forest" seems generous, especially as other ecosystems might be expected to have very different ET (e.g. crops) and would therefore alter estimates of Sr to a large extent. It might be helpful to see the distribution of fractional forest cover present in your "forest" pixels or to otherwise show that the most of the area you are analysing is more fully forested than 50% coverage.
**Response:** It is widely accepted to use a threshold exceeding 50% to differentiate natural forests, characterised by woody vegetation cover greater than 50%, from savannas, where woody vegetation cover is 50% or less (Staal et al. 2018; Zemp et al. 2017). However, we acknowledge that in some instances, evaporation from non-forest land use could affect the overall evaporative trends of a pixel, especially as we aggregate the dataset to a coarser resolution.

Following the reviewer's suggestion, we present data on the fraction of forest cover within a pixel (*see figure below*). Our analysis indicates that instances of forest pixels containing land uses typically associated with non-forest areas frequently arise at the interface of natural and human-influenced regions. However, a visual comparison of these forested areas with the transition zones identified in our study (Fig. 3) reveals that they do not influence the overall findings significantly.

[Figure]

Forest defined by Globcover at 300m resolution

[Figure]

Forest after interpolation to 0.25° resolution
*Fraction of forest within a pixel*

[Figure]

50%                    100%

Since this response sheet will be made publicly available along with the article, we have not made any changes to the revised manuscript regarding this comment.

*Staal, Arie, et al. "Forest-rainfall cascades buffer against drought across the Amazon." Nature Climate Change 8.6 (2018): 539-543.*

*Zemp, Delphine Clara, et al. "Self-amplified Amazon forest loss due to vegetation- atmosphere feedbacks." Nature communications 8.1 (2017): 14681.*

7. The model agreement threshold of 20% seems too low for gaining a robust understanding of likely future transitions. This is especially concerning as the area with >50% model agreement is so small (for example in Figure 2, the forest-savanna transition in Africa).

*Response:* While a threshold of >20% may seem low given the total number of ESMs analysed, it is important to recognise the variable and often limited capabilities of these ESMs, particularly in simulating biophysical interaction and emerging properties due to our limited understanding of the Earth system (Lenton et al., 2019; Stevens and Bony, 2013). **Opting for a majority-based consensus in ESMs could overlook critical tipping risks identified by a minority of models, which might provide insights as valid as those from more widely agreeing models** (Arora et al., 2023; Reyer et al., 2015). [Pg 9, Ln 282-290]

Recognising these challenges in accurately modelling land surface interactions and transitions within ESM, our study showcases model agreement levels of both >20% and >50% (Fig. 2). Contrary to previous studies that either relied on a single model or used an ensemble of hydroclimate estimates from 6 to 7 ESMs for projecting tipping risks (Table S3), which could introduce a selective bias, our approach seeks to address this concern.

We believe that by considering simulations from multiple ESMs under different SSP scenarios, not only do we highlight the agreements and conflicts between potential transitions; but also allow future studies to disentangle vegetation-climate feedbacks and improve the modelling of local-scale interactions (e.g., vegetation's water-uptake profile, species response to $CO_2$ fertilisation) in the ESMs. [Pg 20, Ln 531-535]

In the revised manuscript, the elements from above-mentioned paragraphs are added to the Methodology subsection '**Projecting forest transitions under future climate change**', and the Discussion subsection '**Limitations**'.

Thank you for bringing this to our attention.

Arora, Vivek K., et al. "Towards an ensemble-based evaluation of land surface models in light of uncertain forcings and observations." Biogeosciences 20.7 (2023): 1313-1355.

Lenton, Timothy M., et al. "Climate tipping points—too risky to bet against." Nature 575.7784 (2019): 592-595.

Reyer, Christopher PO, et al. "Forest resilience and tipping points at different spatio-temporal scales: approaches and challenges." Journal of Ecology 103.1 (2015): 5-15.

Stevens, Bjorn, and Sandrine Bony. "What are climate models missing?." science 340.6136 (2013): 1053-1054.

8. Line 48: The '-' is a typo? This sentence doesn't make sense.
   **Response:** Yes, this is a typo. We appreciate your attention in highlighting this. The sentence has been corrected in the revised manuscript. [Pg 2, Ln 55-58]

9. Typo in Figure 2 panel a, under 'transition to a more water-stressed state" (hydraulic failures).
   **Response:** Thank you for bringing this to our attention. We have revised the text in our figure panel. [Pg 13]

---

## Author Response (AR2)

**Multi-fold increase in rainforests tipping risk beyond 1.5-2°C warming**

**Response to Reviewer #2**

*Overall, I think the reviewers did a great job of addressing the previous comments. In particular, the authors have greatly improved the clarity of their methods explanations. They have also elaborated more fully on the limitations and assumptions of their work, which has strengthened the ability of a reader to contextualize the authors' findings.*

**Response:** We appreciate the reviewer's feedback and are glad that the revisions have appropriately addressed their previous concerns.

*Methods restructuring:*

*The authors add full descriptions of their transition categories as well as an updated appendix with further details on the methods. The changes to the methods and appendix greatly improved my ability to understand the analyses that have been done. I have just two suggestions:*

— *First, in section 2.3.2, it would help to more clearly delineate what information under the headings of "Lowly water-limited forest", "moderately..", "etc.." (sections i, ii, iii, iv) comes from this study vs what comes from the previous Singh study or is "common knowledge" (maybe requiring citations?). For example, Line 218 mentions ecosystem "maximum rooting depths typically between 15-20m." Where does this number come from? Similarly, see line 238, which states "this also suggests that, for savanna, deeper roots don't always necessitate a high Sr". This seems more like a result or from a previous paper also?*

**Response:** All qualifications of tree cover, root zone storage capacity, and maximum rooting depth for different forest-savanna classifications are based on a previous study. We have now appropriately cited Singh et al. (2020) where necessary, with references to potential mechanisms taken from other studies cited separately. [Pg 7-8, Ln 195-235]

— *Second, I find that the Appendix was incredibly helpful. Section A2 clarifies the calculation and interpretation of Sr and greatly enhances the manuscript. Figure A1 is great too! However, I wonder if section A3 (or even both appendices…) could be incorporated into the main text?*

**Response:** We kindly request that the reviewer allow us to retain the Appendices in their current form. In our previous revision, we attempted to incorporate them into the main methods text, but this disrupted the flow of the methodology and made the text unnecessarily dense/complex, potentially leading to confusion. This complexity may have been evident in our original submission, where we didn't use appendices.

Presently, the only way to include the appendices' information in the main text of Methods section without disrupting the flow would be to significantly condense it. However, this would sacrifice the level of detail in our discussion (e.g., the technicalities of root zone storage calculation, subsoil interactions within the scope of this study, how previous studies have projected tipping points and the rationale behind our methodology for analysing such transitions).

Moreover, we believe that retaining the discussion in the appendices is beneficial because, in ESD, the appendices are considered an integral part of the main paper and are appropriately highlighted for readers.

*My biggest hesitation with the methodology originally was exactly what you state here - that Sr cannot actually distinguish between forest and savanna ecosystems on its own. I think your argument here that*

you can use future climate to calculate Sr, and therefore forest and savanna transitions, by assuming equilibrium between future climate and landcover is valid and provides interesting results. However, I wonder if the manuscript would benefit from having this assumption laid out very clearly (as you do here in the appendix) in the methods section. Perhaps somewhere around section 2.3.3, which begins "A recent study..demonstrated that Sr effectively represents an ecosystem's above-ground state.." and seems to directly contradict the appendix note about Sr not being able to fully achieve this. Specifying that in the present, we can relate Sr to observed forest vs savanna patterns and then use assumptions to project those into the future would be helpful.

***Response:*** We agree with the reviewer and have now added the following sentences to the beginning of Section 2.3.3:

*"Due to the lack of appropriate metrics for vegetation structure (e.g., tree cover density, tree height, floristic patterns) and the reliance on assumptions about future land-use change (i.e., prescribed rather than biophysically simulated) in ESMs (Hurtt et al., 2020), we use hydroclimate from ESMs as a proxy to project forest transitions under future climate conditions. Using this proxy, we assume that the hydroclimate projected for the end of the 21$^{st}$ century and the ecosystem are in equilibrium (Staal et al., 2020)."* [Pg 8, Ln 248-252]

Small comments:
− In this new revision, there are quite a few grammatical and spelling errors. I have pointed out a few below, but I suspect there are more that I didn't catch.

***Response***: Thank you for your thorough review of our work and your careful attention to detail in identifying grammatical and spelling errors. We have now carefully reviewed the entire manuscript to address any such mistakes.

− Line 30, "Meanwhile, recovering to a less water-limited state gradually diminishes." This sentence doesn't make sense to me in the flow of the abstract. Could it possibly be deleted?

***Response***: We agree with the reviewer and have removed the sentence from the abstract as suggested.

− Line 33: "their immediate" should just be "the immediate"?

***Response***: Thank you. Changes have been made to the revised manuscript. [Pg 1, Ln 31]

− Line 42: Grammatical fix: "…decrease in precipitation and an increase in seasonality and atmospheric water demand."

***Response***: Thank you for your suggestion. However, we have now revised the whole sentence for clarity: *"However, these forests are under severe pressure from climate and land-use changes (Davidson et al., 2012; Lewis et al., 2015; Malhi et al., 2008). These changes result **in decreased precipitation, increased seasonality, and higher atmospheric water demand** (Malhi et al., 2014), leading to soil moisture deficits that inhibit plant growth (Singh et al., 2020; Wang-Erlandsson et al., 2022)"*. [Pg 2, Ln 39-40]

− Line 53: Shedding leaves, or shredding leaves (as written)?

***Response***: Thank you for pointing that out. The correct term is "shedding leaves," and we have made this correction in the revised manuscript. [Pg 2, Ln 50]

− Line 56: I really liked this sentence - good summary of the goal!

***Response***: Thank you.

– Line 61: Capacity should be plural? The sentence might be clearer if written "This limits the capacity of ESMs to…"

*Response*: Thank you. We have made this correction in the revised manuscript. [Pg 2, Ln 58]

– Line 82: "Yet, forest resilience is often assessed based on changes in forest carbon stocks (Huntingford et al., 2013; Parry et al., 2022) or precipitation (Hirota et al., 2011; Staal et al., 2020; Zemp et al., 2017); and rarely on the subsoil moisture availability of the ecosystem (Singh et al., 2022). " This is a super important point and great motivator for your study.

*Response*: Thank you.

– Line 198-99: Slightly weird grammar here with the "and the potential of them approaching the threshold.." This full sentence could possibly be broken up into multiple, or else everything after the "hydroclimatic conditions on forest ecosystems" could be deleted.

*Response*: Indeed, this part does seem somewhat repetitive, so, following the reviewer's suggestion, we have removed the aforementioned sentences. [Pg 6-7, Ln189-194]

– Line 199-201: Is this sentence "According to Singh…benefits available to ecosystem" necessary? I wonder if it is a bit tangential to the argument of this paragraph which is that Sr can be used to diagnose ecosystem state? Bringing in the (true) fact that plants can also change their Sr might also confuse readers as to what part of this puzzle you are primarily addressing here.

*Response*: Thank you for your observation. We agree that this may have led to confusion about the paragraph's primary focus. To improve clarity, we have revised this paragraph as:

*"A recent study by Singh et al. (2020) demonstrated that $S_r$ can effectively represent an ecosystem's above-ground state (i.e., whether it is a forest or savanna) and its level of water–stress, based on root-zone moisture availability. In this study, we refine their terminology from 'water-stressed state' to 'water-limited state' to more precisely describe the effects of changes in hydroclimatic conditions on forest and savanna ecosystems. They classified these terrestrial ecosystem responses into four distinct categories based on the relationship between tree cover density and root zone storage capacity ($S_r$) (for a more detailed description, see Singh et al., 2020):…".* [Pg 6-7, Ln189-194]

– Line 238: "Completive"? Do you mean complementary? Or complete?

*Response*: Thank you for pointing that out. The correct term is "competitive", and we have made this correction in the revised manuscript. [Pg 8, Ln 233]

– Line 456-460: Suggest breaking into two sentences to lose the semicolon and improve readability.

*Response*: Yes. We have now divided these into two sentences:

*"These climate change-induced shifts in ITCZ can potentially both mitigate and exacerbate the effects of (accumulated) water-deficit on the forest ecosystem, especially critical for highly water-limited forests, even without considering the changes to atmospheric moisture flow caused by localised deforestation (Leite-Filho et al., 2021; Schumacher et al., 2022; Staal et al., 2018; Wunderling et al., 2022). This underscores the importance of including changes in atmospheric circulation in studies that analyse the impact of future climate on the resilience of forest ecosystems (Staal et al., 2020; Zemp et al., 2017)."* [Pg 15, Ln 453-458]

– Line 529: Typo, should be "rely" instead of "reply"?

*Response*: Thank you. Changes have been made to the revised manuscript. [Pg 18, Ln 527]

– Line 539 and 541: Might be clearer to say "projected climate" instead of just climate?
  ***Response***: Thank you. Changes have been made to the revised manuscript. [Pg 18, Ln 537 and Ln 539]

– Line 571: Remove first word of the sentence "although" or add in second half of sentence if it is missing?
  ***Response***: Thank you. Changes have been made to the revised manuscript. [Pg 19, Ln 569]

– Line 665: Missing . at end of sentence
  ***Response***: Thank you. Changes have been made to the revised manuscript. [Pg 24, Ln 663]

---

## Author Response (AR3)

**Multi-fold increase in rainforests tipping risk beyond 1.5-2⁰C warming**

**Response to Editor**

Thanks for your revision. Line 278-279 can be more informative, e.g., To classify savannas under future climate conditions, we assume that the vegetation type is in equilibrium with the climate (see Appendix A3 for detailed steps).

***Response:*** We appreciate the editor's feedback, and have now added the following sentence:

"*To classify savanna under future climate conditions, we assume the ecosystem is in equilibrium with the projected climate (see detailed steps in Appendix A3).*" [Pg 8, Ln 255-256]